# Memorization Capacity for Additive Fine-Tuning with Small ReLU Networks

**Jy-yong Sohn**[*1]  **Dohyun Kwon**[*2,3]  **Seoyeon An**[1]  **Kangwook Lee**[4]

[1]Department of Statistics and Data Science, Yonsei University, Republic of Korea
[2]Department of Mathematics, University of Seoul, Republic of Korea
[3]Center for AI and Natural Sciences, Korea Institute for Advanced Study, Republic of Korea
[4]Department of Electrical and Computer Engineering, University of Wisconsin-Madison, WI, USA

## Abstract

Fine-tuning large pre-trained models is a common practice in machine learning applications, yet its mathematical analysis remains largely unexplored. In this paper, we study fine-tuning through the lens of memorization capacity. Our new measure, the Fine-Tuning Capacity (FTC), is defined as the maximum number of samples a neural network can fine-tune, or equivalently, as the minimum number of neurons $(m)$ needed to arbitrarily change $N$ labels among $K$ samples considered in the fine-tuning process. In essence, FTC extends the memorization capacity concept to the fine-tuning scenario. We analyze FTC for the *additive* fine-tuning scenario where the fine-tuned network is defined as the summation of the frozen pre-trained network $f$ and a neural network $g$ (with $m$ neurons) designed for fine-tuning. When $g$ is a ReLU network with either 2 or 3 layers, we obtain tight upper and lower bounds on FTC; we show that $N$ samples can be fine-tuned with $m = \Theta(N)$ neurons for 2-layer networks, and with $m = \Theta(\sqrt{N})$ neurons for 3-layer networks, no matter how large $K$ is. Our results recover the known memorization capacity results when $N = K$ as a special case.

## 1 INTRODUCTION

As a branch of machine learning theory, the expressive power of neural networks is investigated for several decades. By using the concept of universal approximation, it is shown that neural networks can approximate a large classes of functions, either in the depth-bounded scenarios [Cybenko, 1989, Funahashi, 1989, Hornik et al., 1989, Barron, 1993] or width-bounded scenarios [Lu et al., 2017, Hanin and Sellke, 2017, Kidger and Lyons, 2020, Park et al., 2020]. Another line of research focused on the memorization ca-

pacity of neural networks [Baum, 1988, Huang and Babri, 1998, Huang, 2003, Yun et al., 2019, Vershynin, 2020, Rajput et al., 2021, Vardi et al., 2021], exploring the capability of neural networks for memorizing finite samples.

Meanwhile, with the advent of large language models [Brown et al., 2020, OpenAI, 2023, Ouyang et al., 2022, Chowdhery et al., 2022, Zhang et al., 2022, Touvron et al., 2023] and foundation models [Bommasani et al., 2021, Radford et al., 2021, Ramesh et al., 2022], the paradigm of pre-training followed by fine-tuning is dominating the machine learning communities. Various empirical results show that a gigantic model pre-trained on large amount of data can be easily fine-tuned to perform well on downstream tasks, given only a small amount of additional data for the target task. Compared with the extensive empirical results, mathematical analysis on fine-tuning large pre-trained models remains largely unexplored.

In this paper, we take the first step in understanding the fine-tunability of pre-trained networks through the lens of memorization capacity. We focus on the scenario where we fine-tune a pre-trained neural network $f$ on dataset $D = \{(\boldsymbol{x}_i, y_i)\}_{i=1}^{K}$ with $K$ samples; here, $\boldsymbol{x}_i \in \mathbb{R}^d$ and $y_i \in \mathbb{R}$ for all $i \in [K]$ where $[K] = \{1, 2, \cdots, K\}$, and we assume $\boldsymbol{x}_i \neq \boldsymbol{x}_j$ for all $i \neq j$. Let $T := \{i \in [K] : f(\boldsymbol{x}_i) \neq y_i\}$ be the set of indices of samples that the pre-trained network $f$ does not fit. The cardinality of this set is denoted by $N := |T| \leq K$. In other words,

$$f(\boldsymbol{x}_i) = y_i \qquad (1)$$

holds for all $i \in [K] \setminus T$, while not guaranteed for $i \in T$. Our aim is to add a neural network $g_\theta$ (parameterized by $\theta$) to the pre-trained network $f$ in a way that the fine-tuned network $f + g_\theta$ satisfies

$$(f + g_\theta)(\boldsymbol{x}_i) = y_i, \quad \forall i \in [K]. \qquad (2)$$

See Fig. 1 for the visualization of the *additive* fine-tuning scenario we focus on. This scenario is motivated by recently proposed additive fine-tuning methods [Zhang et al., 2020,

---

[*]Equal Contribution

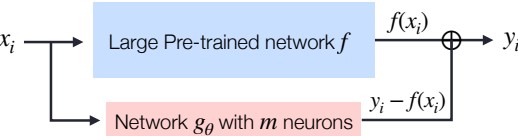

Figure 1: Additive fine-tuning scenario where the pre-trained network $f$ is fine-tuned to $f + g_\theta$, in order to fit the dataset $D = \{(\boldsymbol{x}_i, y_i)\}_{i=1}^K$. Here, the pre-trained network already fits $N$ samples $\{(\boldsymbol{x}_i, y_i)\}_{i \in [K] \setminus T}$, where $T \subseteq [K]$ is the set of indices where $y_i \neq f(\boldsymbol{x}_i)$. We use $g_\theta$ to fill the gap between $f(\boldsymbol{x}_i)$ and $y_i$, for $i \in T$.

Fu et al., 2021, Cao et al., 2022], and especially, the side-tuning [Zhang et al., 2020] where a side network $g_\theta$ is added to the pre-trained network $f$. Since our model does not cover other popular fine-tuning methods including LoRA [Hu et al., 2021], extending our theoretical results to such popular methods is remained as a future work. Under such setting, we define the fine-tuning capacity (FTC) of a neural network $g_\theta$ as below.

**Definition 1.1** (FTC). *The fine-tuning capacity of a given neural network $g_\theta$ is the maximum number $N$ satisfying the following property: for all $\boldsymbol{x}_i \in \mathbb{R}^d$, $y_i \in \mathbb{R}$, for all $T \subseteq [K]$ satisfying $|T| = N$, and for any choices of function $f$ satisfying $f(\boldsymbol{x}_i) = y_i$ for all $i \in [K] \setminus T$, there exists parameter $\theta$ such that $(f + g_\theta)(\boldsymbol{x}_i) = y_i$ all $i \in [K]$.*

Under such a setting, we establish the upper/lower bounds on FTC, when $g$ is 2-layer ReLU network or 3-layer ReLU network. Our main contributions are summarized below:

- We define a new metric called Fine-Tuning Capacity (FTC), which measures the maximum number of samples $N^\star$ a neural network with $m$ neurons can fine-tune. Equivalently, we define the minimum number of neurons $m^\star$ needed to arbitrarily change $N$ labels among $K$. FTC can be considered as an extension of memorization capacity, tailored for the fine-tuning scenario.

- For 2-layer ReLU networks, we establish tight upper and lower bounds on $m^\star$, in Theorem 4.1. The upper bound is obtained by a novel neural network construction for fine-tuning. Our construction requires less number of neurons than conventional constructions developed in the memorization capacity literature when $K \geq 3N + 2$. By using our bounds on $m^\star$, we also provide an equivalent statement in Corollary 4.2, showing the tight bounds on the fine-tuning capacity $N^\star$.

- For 3-layer ReLU networks, we obtain tight upper and lower bounds on $m^\star$ in Theorem 5.1. Our results imply that $N$ samples can be fine-tuned with $m = \Theta(\sqrt{N})$ neurons without any dependence on $K$. We also provide an equivalent statement in Corollary 5.2, showing the tight upper and lower bounds on $N^\star$.

## 2 RELATED WORKS

**Fine-Tuning** Various methods for efficient fine-tuning are introduced in recent years [Houlsby et al., 2019, Zhang et al., 2020, Zaken et al., 2021, He et al., 2021], which fine-tune only a small part of pre-trained models to adapt it for target tasks. There are some mathematical analysis on fine-tuning [Wu et al., 2022, Zeng and Lee, 2023, Giannou et al., 2023, Englert and Lazic, 2022, Oymak et al., 2023, Du et al., 2020, Malladi et al., 2023] or more broadly on transfer learning [Tripuraneni et al., 2020, Maurer et al., 2016], but none of them analyzed the fine-tunability of large pre-trained models using the lens of memorization capacity.

**Memorization** One concept relevant to FTC is *memorization capacity* which measures the ability of memorizing given feature-label pairs $\{(\boldsymbol{x}_i, y_i)\}_{i=1}^K$. Finding the bounds on the memorization capacity is considered in recent works on various networks [Zhang et al., 2016, Yun et al., 2019, Vershynin, 2020, Rajput et al., 2021, Nguyen and Hein, 2018, Hardt and Ma, 2016, Kim et al., 2023]. The effect of memorization in large language models is explored in recent works, both for pre-training [Carlini et al., 2022, Ippolito et al., 2022] and for fine-tuning [Zeng et al., 2023].

## 3 FINE-TUNING CAPACITY

Note that a notion of additive FTC given in Definition 1.1 contains the pre-trained network $f$, but one can confirm that FTC does not depend on $f$ since $f(\boldsymbol{x}_i) = y_i$ holds for all $i \in [K] \setminus T$, for every pre-trained network $f$ we are considering. Below we provide an equivalent simpler definition.

**Definition 3.1** (FTC, equivalent form). *For a given positive integer $K$, the fine-tuning capacity (FTC) of a given neural network $g_\theta$ is*

$$N_{\text{FTC}}^\star(g, K) := \max_{N \in \{0, 1, \cdots, K\}} N \text{ such that}$$

$$\forall T \subseteq [K] \text{ with } |T| = N, \forall \boldsymbol{x}_i \in \mathbb{R}^d, \forall z_i \in \mathbb{R},$$

$$\exists \theta \text{ satisfying } \begin{cases} g_\theta(\boldsymbol{x}_i) = z_i & \forall i \in T, \\ g_\theta(\boldsymbol{x}_i) = 0 & \forall i \in [K] \setminus T. \end{cases}$$
$$(3)$$

This definition is a generalization of conventional memorization capacity shown below, when the condition is relaxed to a special case, $T = [N]$.

**Definition 3.2** (Memorization Capacity [Yun et al., 2019]). *The memorization capacity of a neural network $g_\theta$ is*

$$N_{\text{MC}}^\star(g) := \max_{N \geq 0} N \text{ such that}$$

$$\forall \boldsymbol{x}_i \in \mathbb{R}^d, \forall z_i \in \mathbb{R}, \exists \theta \text{ with } g_\theta(\boldsymbol{x}_i) = z_i \quad \forall i \in [N] \quad (4)$$

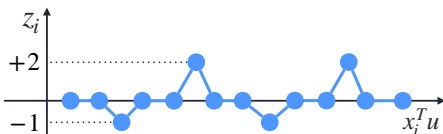

Figure 2: Proving Theorem 4.1 for $K = 14$, $N = 4$.

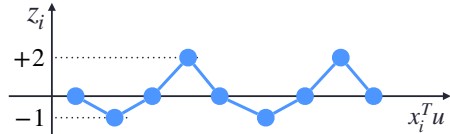

Figure 3: Proving Theorem 4.1 for $K = 9$, $N = 4$.

**Remark 1.** *The memorization capacity and the fine-tuning capacity has a trivial bound: for any neural network $g$ and for arbitrary $K > 0$,*

$$N_{\text{FTC}}^{\star}(g, K) \leq N_{\text{MC}}^{\star}(g). \quad (5)$$

Note that FTC is defined as the maximum number of samples $N$ we can fine-tune using a given network $g$. One can also consider an equivalent definition: the minimum number of neurons $m$ contained in $g$ to successfully fine-tune $N$ samples, which is formally stated below.

**Definition 3.3** (FTC, equivalent form, in terms of # neuron). *The minimum number of neurons required for fine-tuning arbitrary $N$ out of $K$ samples, is defined as*

$$m_{\text{FTC}}^{\star}(N, K) := \min_{m \geq 0} m \text{ such that}$$

$$\forall T \subseteq [K] \text{ with } |T| = N, \forall \boldsymbol{x}_i \in \mathbb{R}^d, \forall z_i \in \mathbb{R},$$
$$\exists \text{ neural network } g_\theta \text{ with } m \text{ neurons satisfying}$$

$$\begin{cases} g_\theta(\boldsymbol{x}_i) = z_i & \text{for all } i \in T, \\ g_\theta(\boldsymbol{x}_i) = 0 & \text{for all } i \in [K] \setminus T. \end{cases}$$

Throughout the paper, we use $N^{\star}$ as a short-hand notation for $N_{\text{FTC}}^{\star}$, and use $m^{\star}$ as a short-hand notation for $m_{\text{FTC}}^{\star}$. Our theoretical results provide bounds on $m^{\star}$ and $N^{\star}$, *e.g.*, Theorem 4.1 is bounding $m^{\star}$, while an equivalent result in Corollary 4.2 is bounding $N^{\star}$.

## 4 FTC OF 2-LAYER FC RELU NETWORKS

A 2-layer fully-connected neural network $g_\theta : \mathbb{R}^d \to \mathbb{R}$ with ReLU activation can be represented as

$$g_\theta(x) = \boldsymbol{W}_2 \sigma(\boldsymbol{W}_1 \boldsymbol{x} + \boldsymbol{b}_1) + \boldsymbol{b}_2, \quad (6)$$

which is parameterized by $\theta = [\boldsymbol{W}_1, \boldsymbol{W}_2, \boldsymbol{b}_1, \boldsymbol{b}_2]$ where $\boldsymbol{W}_1 \in \mathbb{R}^{m \times d}$, $\boldsymbol{W}_2 \in \mathbb{R}^{1 \times m}$, $\boldsymbol{b}_1 \in \mathbb{R}^m$, and $\boldsymbol{b}_2 \in \mathbb{R}$. Here, $m$ is the number of hidden neurons, and $\sigma$ is the ReLU activation. The below result states the bounds on $m$ for 2-layer FC ReLU networks.

**Theorem 4.1.** *Let $K \geq 3$.*

1. *For all $T \subseteq [K]$, $|T| = N$, $\boldsymbol{x}_i \in \mathbb{R}^d$, $z_i \in \mathbb{R}$, $i \in [K]$, there exists a 2-layer fully-connected ReLU network $g$ with $m$ neurons satisfying equation 3 and*

$$m \leq \min\{3N + 1, K - 1\}.$$

2. *For given $T \subseteq [K]$, $|T| = N$, $\boldsymbol{x}_i \in \mathbb{R}^d$, $z_i \in \mathbb{R}$, $i \in [K]$, suppose that equation 3 holds for some 2-layer fully-connected ReLU network $g$ with $m$ neurons. Then,*
$$\min\{3N, K - 2\} \leq m.$$

*Thus,*

$$\min\{3N, K - 2\} \leq m^{\star} \leq \min\{3N + 1, K - 1\}. \quad (7)$$

The proof of this theorem is given in Sec. 4.1 and Sec. 4.2.

**Remark 2.** *If $N = K$ (i.e., fine-tuning changes all labels), then the result of Theorem 4.1 reduces to*

$$m^{\star} + 1 \leq N = K \leq m^{\star} + 2.$$

*The upper bound of $N$ coincides with the upper bound of memorization capacity studied in the result of [Yun et al., 2019]. On the other hand, the lower bound is consistent with the one given in [Zhang et al., 2016].*

Below we state the bounds for the fine-tuning capacity of a 2-layer fully-connected ReLU, directly obtained from the above theorem.

**Corollary 4.2** (FTC of 2-layer FC ReLU). *Suppose $K \geq 3$. For given $m \in \mathbb{N}$, let $N^{\star}$ be the fine-tuning capacity of a 2-layer fully-connected ReLU network $g$ given in equation 6 with $m$ neurons.*

1. *If $K \geq m + 2$, then*

$$\left\lfloor \frac{m-1}{3} \right\rfloor \leq N^{\star} \leq \frac{m}{3}. \quad (8)$$

2. *If $K \leq m + 1$, then $N^{\star} = K$.*

*Proof.* Suppose that $K \geq m + 2$. By Theorem 4.1.1, for $|T| = \lfloor \frac{m-1}{3} \rfloor$, there exists a 2-layer fully-connected ReLU network $g$ where the number of neurons of $g$ is less than or equal to

$$\min\left\{3 \left\lfloor \frac{m-1}{3} \right\rfloor + 1, K - 1\right\} = 3 \left\lfloor \frac{m-1}{3} \right\rfloor + 1 \leq m.$$

Here, $K \geq m + 2$ yields the first equality while the second inequality follows from the definition of the floor function. This yields the lower bound of equation 8. On the other hand, due to Theorem 4.1.2 and $K \geq m + 2$, we conclude the upper bound of equation 8.

Suppose that $K \le m + 1$. Similarly, by Theorem 4.1.1 for $|T| = K$, there exists a 2-layer fully-connected ReLU network $g$ where (the number of neurons of $g$) $\le K - 1 \le m$. This yields $N \ge K$. As $N \le K$ always holds due to our construction, we conclude $N = K$. $\qquad\square$

In other words, for sufficiently large $K$, the fine-tuning capacity $N$ does not depend on the size $K$ of the underlying dataset $D$ but the number of neurons $m$.

### 4.1 PROOF OF LOWER BOUND ON $m^\star$

We first prove the lower bound in Theorem 4.1 when $K \ge 3N + 2$. Define $T = \{3, 6, 9, ..., 3N\}$, and define $\boldsymbol{x}_i = i\boldsymbol{u}$ for all $i \in [N]$ for arbitrary vector $\boldsymbol{u} \in \mathbb{R}^d$. Let $z_i = 2$ if $i = 6k$ for some $k \in \mathbb{N}$ and $z_i = -1$ if $i = 6k - 3$ for some $k \in \mathbb{N}$. See Fig. 2 when $K = 14$ and $N = 4$. Then, $\bar{g}_\theta(t) := g_\theta(t\boldsymbol{u})$ is a piecewise affine function with at least $3N + 1$ pieces. Recall that using Theorem 3.3 of [Yun et al., 2019] for 2-layer ReLU network, $\bar{g}_\theta(t)$ is having $m + 1$ pieces, since ReLU activation is a piecewise linear function with two pieces. Thus, $m \ge 3N$ holds. Note that given $K$ and $N$, the above proof scheme specifies $T, \{\boldsymbol{x}_i, z_i\}_{i=1}^K$ and counts the number of pieces for the piecewise-linear function $\bar{g}_\theta(t) = g_\theta(t\boldsymbol{u})$, under the setting of $\boldsymbol{x}_i = i\boldsymbol{u}$.

We use a similar technique for proving the lower bound on $m$ when $N \le K < 3N + 2$. Consider revising $(T, x_i, z_i)$ triplet (defined for $K \ge 3N + 2$ case), so that $K < 3N + 2$ condition is satisfied. For example, when $K = 2N + 1$, the triplet is revised as $T = \{2, 4, \cdots, 2N\}$, $z_i = 2$ if $i = 4k$ for some $k \in \mathbb{N}$ and $z_i = -1$ if $i = 4k - 2$ for some $k \in \mathbb{N}$, as illustrated in Fig. 3, which has $K - 1 = 8$ pieces. For general $K$ and $N$ satisfying $K < 3N + 2$, one can choose $(T, x_i, z_i)$ triplet such that the corresponding $\bar{g}_\theta(t) = g_\theta(t\boldsymbol{u})$ has $K - 1$ pieces. Thus, the number of pieces $p(K)$ of $\bar{g}_\theta(t)$ we constructed can be represented as

$$p(K) = \begin{cases} 3N + 1, & \text{if } K \ge 3N + 2 \\ K - 1, & \text{if } N \le K < 3N + 2 \end{cases} \quad (9)$$

This completes the proof.

### 4.2 PROOF OF UPPER BOUND ON $m^\star$

We here establish the upper bound in Theorem 4.1. To be specific, in Theorem 4.4, we establish the upper bound on $m^\star$ in terms of the partition of $[K] \setminus T$. The key idea is to remove all points of $[K] \setminus T$ except for the endpoints of each block as in Figure 4. More discussion will be provided below.

Let us introduce some terminology. For a given set $I$, a partition $\mathcal{P}$ of $I$ is a set of nonempty subsets $P$ of $I$ such that every element in $I$ is in exactly one of these subsets. We denote $P \in \mathcal{P}$ by the *block* of $\mathcal{P}$.

**Definition 4.3.** *For $I \subset [K]$, we say that $\mathcal{P}(I)$ is the consecutive partition of $I$ if all consecutive integers in $I$ are*

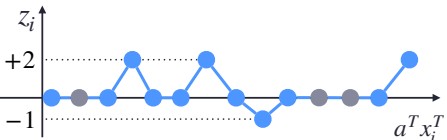

Figure 4: Illustration of the neural network constructed in Theorem 4.4 with $K = 14$ and $T = \{4, 7, 9, 14\}$. $\mathcal{P}([K] \setminus T)$ is the partition $\mathcal{P}(I)$ given in Example 1. The gray points are the removed ones.

*included in the same block, i.e., for $i, j \in I$, $i$ and $j$ are in the same block of $\mathcal{P}(I)$ if and only if $|i - j| = 1$.*

**Example 1.** *If $I = \{1, 2, 3, 5, 6, 8, 10, 11, 12, 13\}$, then the consecutive partition is defined as*

$$\mathcal{P}(I) = \{\{1, 2, 3\}, \{5, 6\}, \{8\}, \{10, 11, 12, 13\}\}, \quad (10)$$

*and the blocks of $\mathcal{P}(I)$ are $\{1, 2, 3\}, \{5, 6\}$, $\{8\}$, and $\{10, 11, 12, 13\}$.*

The following theorem shows that the upper bound of $m$ is given in terms of the size of the blocks in $\mathcal{P}([K] \setminus T)$. For proving Theorem 4.1, we find the uniform bound for general datasets using this bound and Lemma 4.7.

**Theorem 4.4.** *Consider the same setting as in Theorem 4.1, and suppose that*

$$\boldsymbol{a}^T \boldsymbol{x}_1 < \boldsymbol{a}^T \boldsymbol{x}_2 < \cdots < \boldsymbol{a}^T \boldsymbol{x}_K \quad (11)$$

*holds for some $\boldsymbol{a} \in \mathbb{R}^d$. Then, there exists an $m$-neuron network $g_\theta(x) = \boldsymbol{W}_2 \sigma(\boldsymbol{1}_m \boldsymbol{a}^T \boldsymbol{x} + \boldsymbol{b}_1) + \boldsymbol{b}_2$, such that equation 3 holds, and*

$$m^\star \le K - 1 - \sum_{P \in \mathcal{P}([K] \setminus T)} \max\{|P| - 2, 0\} \quad (12)$$

*Here, $\boldsymbol{W}_2 \in \mathbb{R}^{1 \times m}$, $\boldsymbol{b}_1 \in \mathbb{R}^m$, and $\boldsymbol{b}_2 \in \mathbb{R}$.*

**Remark 3.** *To minimize the width $m$ of the network, a smaller number of blocks of bigger sizes would be ideal. If we only have one block in the partition $\mathcal{P}([K] \setminus T)$, then Eq. 12 is given as*

$$m^\star \le K - 1 - |[K] \setminus T| + 2 = N + 1.$$

*This happens when $\boldsymbol{a}^T \boldsymbol{x}_i s$ for $i \in [K] \setminus T$ are segregated for some $\boldsymbol{a}$, e.g., when $[K] \setminus T = \{i, i + 1, \cdots, j\}$ for some $i < j$ in $[K]$. Thus, some appropriate projection yields a smaller number of neurons required.*

**Remark 4.** *The worst scenario is when every block of $\mathcal{P}([K] \setminus T)$ has 2 or less elements, as in Figures 2 and 3. Note that if $K - N$ is much larger than $N$, then this scenario cannot occur. This is because the number of blocks cannot be larger than $\min\{K - N, N\} + 1$ from Lemma 4.6.*

*Proof of Theorem 4.4.* First, we consider the case when

$$|P| \leq 2 \text{ for all } P \in \mathcal{P}([K] \setminus T). \tag{13}$$

In this case, it suffices to prove $m \leq K$. As $K$ neurons can represent $K$ data points, the inequality directly follows from the standard argument as in Zhang et al. [2016] and also shown in Lemma 4.5.

Let us consider general cases. The main strategy is to remove data points so that equation 13 holds. Specifically, except for two endpoints of each block $P \in \mathcal{P}([K] \setminus T)$, we remove $i \in [K] \setminus T$ from $[K]$. Let us denote this new subset of $[K]$ by

$$J = \{j_1 < j_2 < \cdots < j_{|J|}\} \tag{14}$$

For example, when $K = 14$ and $T = \{4, 7, 9, 14\}$ as in Fig. 4, the consecutive partition $\mathcal{P}(I)$ is given in Eq. 10, and thus we remove $\{2, 11, 12\}$ from $[K] = \{1, 2, \cdots, 14\}$, which gives us $J = \{1, 3, 4, 5, 6, 7, 8, 9, 10, 13, 14\}$. Note that the number of data points in $J$ is

$$|J| := K - \sum_{P \in \mathcal{P}([K] \setminus T)} \max\{|P| - 2, 0\}. \tag{15}$$

Applying Lemma 4.5 with $A := \{(\boldsymbol{a}^T \boldsymbol{x}_j, \boldsymbol{z}_j)\}_{j \in J}$ and $m = |J| - 1$, there exist $\boldsymbol{W}_2 \in \mathbb{R}^{1 \times m}$, $\boldsymbol{b}_1 \in \mathbb{R}^m$, and $\boldsymbol{b}_2 \in \mathbb{R}$ such that

$$h_\theta(\boldsymbol{a}^T \boldsymbol{x}_j) = \boldsymbol{W}_2 \sigma(\boldsymbol{1}_m \boldsymbol{a}^T \boldsymbol{x}_j + \boldsymbol{b}_1) + \boldsymbol{b}_2 = z_j$$

for all $j \in J$. Thanks to $h_\theta(\boldsymbol{a}^T \boldsymbol{x}) = g_\theta(\boldsymbol{x})$, we conclude $g_\theta(\boldsymbol{x}_j) = z_j$ for all $j \in J$.

Lastly, for $i \in [K] \setminus J$, there exist two endpoints $j_i < j_{i+1}$ such that $[j_i, j_{i+1}] \cap ([K] \setminus T)$ in $\mathcal{P}([K] \setminus T)$, and $j_i < i < j_{i+1}$. Note that $h_\theta$ constructed in Lemma 4.5 is linear in $[\boldsymbol{a}^T \boldsymbol{x}_{j_i}, \boldsymbol{a}^T \boldsymbol{x}_{j_{i+1}}]$. Since $g_\theta(\boldsymbol{x}_{j_i}) = g_\theta(\boldsymbol{x}_{j_{i+1}}) = 0$, we conclude that $g_\theta(\boldsymbol{x}_i) = 0$ as desired. $\square$

**Lemma 4.5.** *For $m \geq 1$ and $A = \{(w_i, z_i)\}_{i=1}^{m+1}$ where $w_1 < w_2 < \cdots < w_{m+1}$, $w_i \in \mathbb{R}$ and $y_i \in \mathbb{R}$. There exist $\boldsymbol{W}_2 \in \mathbb{R}^{1 \times m}$, $\boldsymbol{b}_1 \in \mathbb{R}^m$, and $\boldsymbol{b}_2 \in \mathbb{R}$ such that*

$$h_\theta(x) = \frac{z_i - z_{i+1}}{w_i - w_{i+1}}(x - w_i) + z_i \tag{16}$$

*for $x \in [w_i, w_{i+1}]$, $i = 1, 2, \cdots, m - 1$ and equation 16 with $i = m$ holds in $[w_m, \infty)$ where $h_\theta(x) = \boldsymbol{W}_2 \sigma(\boldsymbol{1}_m x + \boldsymbol{b}_1) + \boldsymbol{b}_2$.*

*Proof.* We prove this by induction. For $m = 1$, choose $\boldsymbol{b}_1 = -w_1$ and $b_1 = z_1$, then $h_\theta(w_1) = z_1$. Take $W_2 = \frac{z_1 - z_2}{w_1 - w_2}$ and we get equation 16 with $i = 1$ in $[w_1, \infty)$.

Suppose that the above result holds for $m = k$. Then, there exist $\boldsymbol{W}_2 \in \mathbb{R}^{1 \times k}$, $\boldsymbol{b}_1 \in \mathbb{R}^k$, and $\boldsymbol{b}_2 \in \mathbb{R}$ satisfying equation 16 in $[w_i, w_{i+1}]$ for $i = 1, 2, \cdots k - 1$ and $[w_k, \infty)$. Using this, we choose $\widetilde{\boldsymbol{W}_2} = (\boldsymbol{W}_2, \frac{z_{k+1} - z_{k+2}}{w_{k+1} - w_{k+2}})$ and $\widetilde{b}_1 = (b_1, -w_{k+1})$. Then, $\widetilde{h_\theta}(x) = \boldsymbol{W}_2 \sigma(\boldsymbol{1}_m x + \boldsymbol{b}_1) + \boldsymbol{b}_2$ satisfies equation 16 with $i = k + 1$ in $[w_{k+1}, \infty)$. $\square$

**Lemma 4.6.** *For $I \subset [K]$ and the consecutive partition, $\mathcal{P}(I)$, it holds that*

$$|\mathcal{P}(I)| \leq \min\{|I|, |[K] \setminus I| + 1\}.$$

*Proof.* First, $|\mathcal{P}(I)| \leq |I|$ directly follows from the definition of the partition.

On the other hand, let $a_i$ and $b_i$ be the end point of each block in $\mathcal{P}([K] \setminus I)$:

$$\mathcal{P}([K] \setminus I) = \{[a_i, b_i] \cap [K] : 1 \leq i \leq |[K] \setminus I|\}.$$

Then,

$$\mathcal{P}(I) = \{[b_i + 1, a_{i+1} - 1] \cap [K] : 1 \leq i \leq |[K] \setminus I| - 1\}$$
$$\cup \{[1, b_i - 1] \cap [K], [a_{[K] \setminus I} + 1, K] \cap [K]\}$$

and thus we conclude that $|\mathcal{P}(I)| \leq |[K] \setminus I| + 1$. $\square$

**Lemma 4.7.** *Under the same setting as in Theorem 4.4, we have*

$$\sum_{P \in \mathcal{P}([K] \setminus T)} \max\{|P| - 2, 0\} \geq \max\{K - 3N - 2, 0\}.$$

*In particular, $J$ given in equation 15 satisfies*

$$|J| \leq \min\{3N + 2, K\}.$$

*Proof.* Recall that $\sum_{P \in \mathcal{P}([K] \setminus T)} |P| = |[K] \setminus T| = K - N$. Using this, we have

$$\sum_{P \in \mathcal{P}([K] \setminus T)} \max\{|P| - 2, 0\} \geq \sum_{P \in \mathcal{P}([K] \setminus T)} (|P| - 2)$$
$$\geq K - N - 2|\mathcal{P}([K] \setminus T)|.$$

By Lemma 4.6, the number of blocks in the partition cannot be larger than $\min\{K - N, N + 1\}$. $\square$

**Proof of upper bound in Theorem 4.1:** Since $\boldsymbol{x}_i \neq \boldsymbol{x}_j$ for all $i \neq j$, there exists $\boldsymbol{a} \in \mathbb{R}^d$ satisfying $\boldsymbol{x}_i^T \boldsymbol{a} \neq \boldsymbol{x}_j^T \boldsymbol{a}$ all $i \neq j$. Without the loss of generality, we assume that equation 11 holds. Then, using Theorem 4.4 and choosing $\boldsymbol{W}_1 = \boldsymbol{1}_m \boldsymbol{a}^T$, there exists

$$g_\theta(\boldsymbol{x}) = \boldsymbol{W}_2 \sigma(\boldsymbol{W}_1 \boldsymbol{x} + \boldsymbol{b}_1) + \boldsymbol{b}_2,$$

satisfying equation 3 and

$$m \leq K - 1 - \sum_{P \in \mathcal{P}([K] \setminus T)} \max\{|P| - 2, 0\}.$$

By Lemma 4.7, we conclude that $m \leq K - 1 - \max\{K - 3N - 2, 0\} = \min\{3N + 1, K - 1\}$.

# 5 FTC OF 3-LAYER RELU NETWORK

Now we analyze FTC of 3-layer fully-connected neural network $g_\theta : \mathbb{R}^d \to \mathbb{R}$ with ReLU activation. Note that 3-layer network can be represented as

$$g_\theta(x) = W_3 \sigma(W_2 \sigma(W_1 x + b_1) + b_2) + b_3. \quad (17)$$

As before, $\sigma$ is the ReLU activation and the network is parameterized by $\theta = [W_1, W_2, W_3, b_1, b_2, b_3]$ where $v \in \mathbb{R}^{d_2}$, $W_1 \in \mathbb{R}^{d_1 \times d}$, $W_2 \in \mathbb{R}^{d_2 \times d_1}$, $W_3 \in \mathbb{R}^{1 \times d_2}$, $b_1 \in \mathbb{R}^{d_1}$, $b_2 \in \mathbb{R}^{d_2}$ and $b_3 \in \mathbb{R}$. Following the setting considered in the memorization capacity of 3-layer neural network [Yun et al., 2019], we consider the scenario when $z_i \in [-1, +1]$ for all $i \in [K]$. For notational simplicity, we denote $z_i = 0$ for $i \in [K] \setminus T$. Below theorem summarizes the upper/lower bound on FTC of 3-layer network.

**Theorem 5.1** (FTC of 3-layer FC ReLU). *Let $K \geq 3$, $T \subseteq [K]$, $|T| = N$, and $g$ be a 3-layer FC ReLU network with $m$ neurons.*

1. *For all $x_i \in \mathbb{R}^d$, $z_i \in \mathbb{R}$, $i \in [K]$, there exists $g$ with $m$ neurons satisfying equation 3 where*

$$m \leq \min\{2\sqrt{K} + \min\{2\sqrt{K}, 3N\}, 6\sqrt{3N+2}\}.$$

2. *For given $x_i \in \mathbb{R}^d$, $z_i \in \mathbb{R}$, $i \in [K]$, suppose that equation 3 holds for some $g$ with $m$ neurons. Then,*

$$\sqrt{2\min\{3N, K-2\} + \frac{1}{4}} - \frac{1}{2} \leq m.$$

*Thus, the minimum number of neurons $m^\star$ is bounded as*

$$\sqrt{2\min\{3N, K-2\} + \frac{1}{4}} - \frac{1}{2}$$
$$\leq m^\star \leq \min\{2\sqrt{K} + \min\{2\sqrt{K}, 3N\}, 6\sqrt{3N+2}\}$$

**Remark 5.** *The upper bound in Theorem 5.1 directly shows that $m^\star \leq \Theta(\sqrt{N})$. The lower bound in Theorem 5.1 indicates two facts:*

- *If $3N \leq K - 2$, then $m^\star \geq \Theta(\sqrt{N})$,*
- *If $3N > K - 2$, then $m^\star \geq \Theta(\sqrt{K})$. Combining this with $K \geq N$, we have $m^\star \geq \Theta(\sqrt{N})$.*

*All in all, $m^\star = \Theta(\sqrt{N})$.*

**Corollary 5.2** (FTC of 3-layer FC ReLU). *For given $m \in \mathbb{N}$, let $N^\star$ be the fine-tuning capacity of a 3-layer fully-connected ReLU network $g$ given in equation 17 with $m$ neurons. Then, $N^\star$ is bounded as below, for different range of $K$:*

1. *If $K \leq \left\lfloor \frac{m^2}{16} \right\rfloor$, then*

$$N^\star = K. \quad (18)$$

2. *If $\left\lfloor \frac{m^2}{16} \right\rfloor + 1 \leq K < \frac{m^2+m+4}{2}$, then*

$$\left\lfloor \frac{m^2}{108} - \frac{2}{3} \right\rfloor \leq N^\star \leq K. \quad (19)$$

3. *If $K \geq \frac{m^2+m+4}{2}$, then*

$$\left\lfloor \frac{m^2}{108} - \frac{2}{3} \right\rfloor \leq N^\star \leq \frac{m^2+m}{6}. \quad (20)$$

*Proof.* Suppose $K \leq \left\lfloor \frac{m^2}{16} \right\rfloor$. By Theorem 5.1.1, for $|T| = K$, there exists a 3-layer fully-connected ReLU network $g$ where the number of neurons of $g$ is less than or equal to

$$\min\{4\sqrt{K}, 2\sqrt{K} + 3K, 6\sqrt{3K+2}\} = 4\sqrt{K} \leq m.$$

This implies that $N \geq K$, and since $N \leq K$ always holds, we can conclude $N^\star = K$.

Suppose $K \geq \left\lfloor \frac{m^2}{16} \right\rfloor + 1$. By Theorem 5.1.1, for $|T| = \left\lfloor \frac{m^2}{108} - \frac{2}{3} \right\rfloor$, there exists a 3-layer fully-connected ReLU network $g$ where the number of neurons of $g$ is less than or equal to

$$\min\left\{4\sqrt{K}, 6\sqrt{3\left\lfloor \frac{m^2}{108} - \frac{2}{3} \right\rfloor + 2}\right\} \quad (21)$$

$$= 6\sqrt{3\left\lfloor \frac{m^2}{108} - \frac{2}{3} \right\rfloor + 2} \leq m, \quad (22)$$

where the inequality is derived from the fact that $4\sqrt{K} \geq m$ for $K \geq \left\lfloor \frac{m^2}{16} \right\rfloor + 1$, and $6\sqrt{3\left\lfloor \frac{m^2}{108} - \frac{2}{3} \right\rfloor + 2} \leq 6\sqrt{3\left(\frac{m^2}{108} - \frac{2}{3}\right) + 2} = m$. Now we can conclude that $N^\star \geq \left\lfloor \frac{m^2}{108} - \frac{2}{3} \right\rfloor$.

Now we derive the upper bound on $N^\star$. If $K \geq \frac{m^2+m+4}{2}$, then

$$m \geq \sqrt{2\min\{3N, K-2\} + \frac{1}{4}} - \frac{1}{2}$$
$$\geq \sqrt{\min\{6N, m^2+m\} + \frac{1}{4}} - \frac{1}{2}$$
$$= \min\left\{\sqrt{6N + \frac{1}{4}}, m + \frac{1}{2}\right\} - \frac{1}{2}$$
$$= \min\left\{\sqrt{6N + \frac{1}{4}} - \frac{1}{2}, m\right\},$$

where the first inequality is from Theorem 5.1.2, and the second inequality is from $K \geq \frac{m^2+m+4}{2}$. We can simplify the above inequalities as $m \geq \sqrt{6N + \frac{1}{4}} - \frac{1}{2}$. Since this holds for any number of fine-tunable samples $N$, we have $N^\star \leq \frac{m^2+m}{6}$.

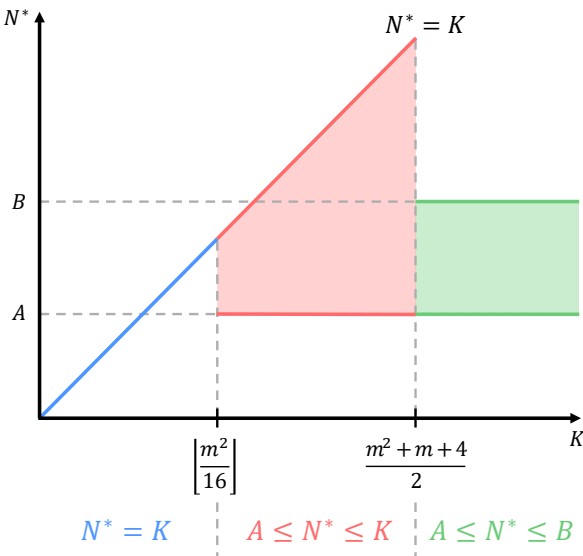

$N^* = K$ (blue)
$A \leq N^* \leq K$
$A \leq N^* \leq B$

Figure 5: Visualization of FTC for 3-layer network in corollary 5.2. In this figure, $A = \left\lfloor \frac{m^2}{108} - \frac{2}{3} \right\rfloor$ and $B = \frac{m^2+m}{6}$.

If $\left\lfloor \frac{m^2}{16} \right\rfloor + 1 \leq K < \frac{m^2+m+4}{2}$, we use a trivial upper bound $N^* \leq K$, which completes our proof. □

Fig. 5 illustrates the results in Corollary 5.2. For different ranges of $K$, we have either a constant $N^* = K$, or upper/lower bounds on $N^*$.

## 5.1 PROOF OF LOWER BOUND ON $m$

We follow the proof technique used in section 4.1. Since the number of minimum pieces $p(K)$ provided in Eq. 9 is still valid for 3-layer neural network, we just need to check how many pieces $\bar{g}_\theta(t)$ has. As stated in the proof of Theorem 3.3 of [Yun et al., 2019], $\bar{g}_\theta(t)$ has $2d_1 d_2 + d_2 + 1$ pieces, where $d_1$ and $d_2$ are the number of neurons in layer 1 and 2, respectively. Thus, we have

$$\min\{3N, K-2\} \leq 2d_1 d_2 + d_2 \leq \frac{m^2}{2} + \frac{m}{2},$$

where the last inequality is from the fact that $2d_1 d_2 + d_2$ is having its maximum value when $d_1 = d_2 = m/2$. Reformulating the above inequality with respect to $m$ completes the proof.

## 5.2 PROOF OF UPPER BOUND ON $m$

We can prove $m \leq U$ for an upper bound $U$ by constructing a 3-layer neural network $g_\theta$ having $U$ neurons, which satisfies Eq. 3 for given $N$ and $K$. Below we construct different types of neural networks satisfying the condition, where each construction gives different upper bounds $U_1 = 4\sqrt{K}, U_2 = 2\sqrt{K} + 3N$ and $U_3 = 6\sqrt{3N+2}$. This completes the proof of $m \leq \min\{U_1, U_2, U_3\}$.

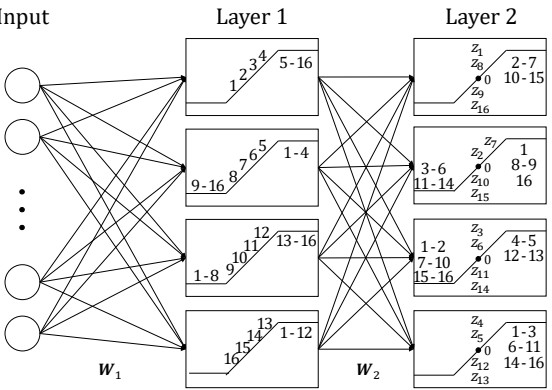

Figure 6: Construction of 3-layer network that achieves upper bound $U_1 = 4\sqrt{K}$ in Sec. 5.2. This construction is directly from [Yun et al., 2019], and each neuron uses hard-tanh activation in Eq. 23. This illustration gives an example when $T = \{7\}$, *i.e.,* we change the label for $\boldsymbol{x}_7$, while maintaining the label for other samples.

**Proof of $m \leq 4\sqrt{K}$:** Recall the neural network $g_\theta$ constructed in the proof of Theorem 3.1 of [Yun et al., 2019], containing $\sqrt{K}$ neurons in both 1st layer and 2nd layer. See Fig. 6 for the illustration of such network, when $K = 16$ and $N = 1$, where the index of fine-tuning sample is $T = \{7\}$. In such case $z_7 \neq 0$ from the definition of $T$. Each box (node) in the figure is a neuron, where the curve inside the box represents the activation function of the neuron. In this figure, each neuron uses hard-tanh activation defined as

$$\sigma_H(x) = \begin{cases} -1, & t \leq -1 \\ t, & -1 < t < +1 \\ +1, & t \geq +1, \end{cases} \quad (23)$$

where $[-1, 1]$ is the *non-clipping* region of $\sigma_H$, and $[-1, 1]^c$ is the *clipping* region of $\sigma_H$. Note that the digit $i$ in the box represents the location where the feature $\boldsymbol{x}_i$ for the $i$-th sample is mapped to. For example, for the first neuron of layer 1, we have $\alpha_1^1(\boldsymbol{x}_i) \in [-1, 1]$ for $i \in \{1, 2, 3, 4\}$ and $\alpha_1^1(\boldsymbol{x}_i) > 1$ for $i \in \{5, 6, \cdots, 16\}$, where

$$\alpha_j^l(\boldsymbol{x}) = \boldsymbol{W}_{l,j}\boldsymbol{x} + b_{l,j} \quad (24)$$

is the input value of node $j$ in layer $l$, when the input for the network $g_\theta$ is given as $\boldsymbol{x}$. Here, the weight matrix and the bias for layer 1 are denoted by $\boldsymbol{W}_1 = [\boldsymbol{W}_{1,1}^T; \cdots; \boldsymbol{W}_{1,\sqrt{K}}^T]$ and $\boldsymbol{b}_1 = [b_{1,1}, \cdots, b_{1,\sqrt{K}}]$, respectively. Given target $\{z_i\}_{i=1}^K$, the proof of Theorem 3.1 of [Yun et al., 2019] specified parameters $\theta = [\boldsymbol{W}_1, \boldsymbol{W}_2, \boldsymbol{W}_3, \boldsymbol{b}_1, \boldsymbol{b}_2, \boldsymbol{b}_3]$ satisfying $g_\theta(\boldsymbol{x}_i) = z_i$ for all $i \in [K]$. Assigning $z_i = 0$ for $i \in [K] \setminus T$ and reusing these parameters is enough to satisfy the desired condition for fine-tuning in Eq. 3. Note that this network uses $2\sqrt{K}$ neurons with hard-tanh activations, which can be converted to a ReLU neural network with $4\sqrt{K}$ neurons, using the fact that one hard-tanh neuron can be expressed with two ReLU neurons. This directly proves

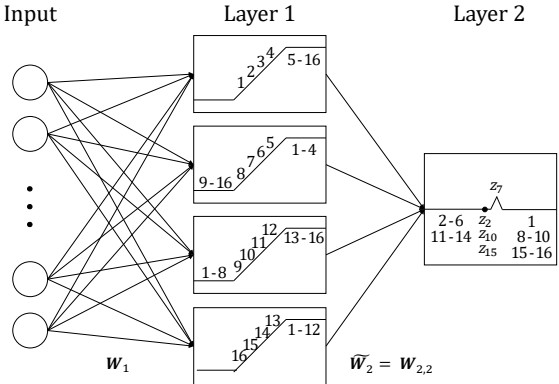

Figure 7: Construction of 3-layer network that achieves upper bound $U_2 = 2\sqrt{K} + 3N$ in Sec. 5.2. This illustration gives an example when $T = \{7\}$.

that $4\sqrt{K}$ ReLU neurons are sufficient for changing the labels of $N \leq K$ samples.

**Proof of $m \leq 2\sqrt{K} + 3N$:** We construct a 3-layer neural network with $U_2 = 2\sqrt{K} + 3N$ neurons which successfully fine-tunes $N$ samples. Fig. 7 illustrates the example of such construction when $K = 16$ and $N = 1$. Here, we assumed $T = \{7\}$, *i.e.,* the label for 7-th sample is fine-tuned, but similar proof can be applied to arbitrary $T$ with $|T| = N = 1$. Here, our goal is to construct $g_\theta$ satisfying $g_\theta(\boldsymbol{x}_7) = z_7 \neq 0$ and $g_\theta(\boldsymbol{x}_i) = 0$ for all $i \neq 7$. Our basic idea is, to follow the construction in Fig. 6, except the activation used in layer 2. The activation in layer 2 is defined as

$$\sigma_B(t) = \begin{cases} 0, & t \notin [z_7 - \delta, z_7 + \delta] \\ \frac{z_7}{\delta}(t - z_7 + \delta), & t \in [z_7 - \delta, z_7) \\ -\frac{z_7}{\delta}(t - z_7 - \delta), & t \in [z_7, z_7 + \delta) \end{cases}$$

for arbitrary $\delta < \min_{i \neq j} \frac{|z_i - z_j|}{2}$, which is having a small bump near $z_7$ as in Fig. 7. Among $\sqrt{K}$ hard-tanh neurons in layer 2 in Fig. 6, we only choose the neuron containing $z_7$ (the non-zero target label) in the non-clipping region of the activation, *e.g.,* the second neuron of layer 2. Given that $\boldsymbol{W}_2$ is the weight matrix for 2nd layer in the construction of Fig. 6, the weight matrix for 2nd layer in the construction of Fig. 7 is defined as $\widetilde{\boldsymbol{W}_2} := \boldsymbol{W}_{2,2}$, the 2nd row of $\boldsymbol{W}_2$. Then, the overall network looks like $g_\theta(\boldsymbol{x}) = \sigma_B(\boldsymbol{W}_{2,2}\sigma_H(\boldsymbol{W}_1\boldsymbol{x} + \boldsymbol{b}_1) + \boldsymbol{b}_2)$, which satisfies $g_\theta(\boldsymbol{x}_7) = z_7$ and $g_\theta(\boldsymbol{x}_i) = 0$ for all $i \neq 7$. Note that the activation $\sigma_B$ with 4 pieces can be constructed by 3 ReLU neurons. Consider constructing $g_\theta$ by adding new neurons (with $\sigma_B$ activation) in layer 2 for each sample we want to fine-tune, which allocates total $3N$ ReLU neurons in layer 2. Since layer 1 (as in Fig. 6) contains $2\sqrt{K}$ ReLU neurons, one can confirm that our construction contains $2\sqrt{K} + 3N$ ReLU neurons.

**Proof of $m \leq 6\sqrt{3N + 2}$:** It is worth nothing that some techniques developed for showing $m \leq 4N + 4$ (shown in

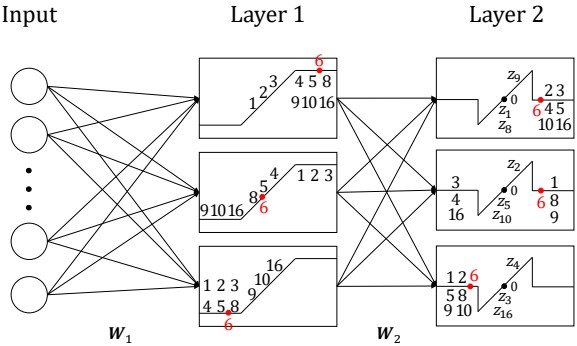

Figure 8: The construction of 3-layer network achieving the upper bound $U_3 = 6\sqrt{3N + 2}$ in Sec. 5.2. Here, we consider the case when $K = 16$, $N = 3$ and $T = \{2, 4, 9\}$.

Appendix) are used to prove $m \leq 6\sqrt{3N + 2}$. Thus, we refer to some equations in Appendix during the proof.

We construct a neural network $g_\theta$ with $6\sqrt{3N + 2}$ ReLU neurons, which fine-tunes $N$ samples. For simplicity, here we provide an example construction when $K = 16$, $N = 3$, and $T = \{2, 4, 9\}$. By using the definition of $J$ in Eq. 14, we have $J = \{1, 2, 3, 4, 5, 8, 9, 10, 16\}$. We will now construct a 3-layer neural network $g_\theta$ satisfying

$$g_\theta(\boldsymbol{x}_i) = z_i, \quad i \in J, \tag{25}$$
$$g_\theta(\boldsymbol{x}_i) = 0, \quad i \notin J, \tag{26}$$

which is illustrated in Fig. 8. Note that Eq. 25 can be easily proved by fitting the network using the samples $\boldsymbol{x}_i$ with $i \in J$. In the rest of the proof, we will show that for such $g_\theta$ satisfying Eq. 25, we can prove Eq. 26. As an example, we will only show $g_\theta(\boldsymbol{x}_6) = 0$, but a similar proof technique can be applied to arbitrary $\boldsymbol{x}_i$ with $i \notin J$.

We construct the first layer by only using samples $\boldsymbol{x}_i$ with $i \in J$. We partition the $|J|$ samples into $\sqrt{|J|}$ groups, following the trick used in [Yun et al., 2019] for 3-layer network. Let us denote the first index of $j$-th group as $s_j^{min}$ and the last index of $j$-th group as $s_j^{max}$. In other words, $J$ is decomposed into $\sqrt{|J|}$ groups as $J = \{s_1^{min}, \cdots, s_1^{max}, \} \cup \cdots \cup \{s_{\sqrt{|J|}}^{min}, \cdots, s_{\sqrt{|J|}}^{max}\}$.

Without loss of generality, we can assume that samples are ordered as $\boldsymbol{v}^T\boldsymbol{x}_1 < \boldsymbol{v}^T\boldsymbol{x}_2 < \cdots < \boldsymbol{v}^T\boldsymbol{x}_K$ for some $\boldsymbol{v}$. Let $c_i := \boldsymbol{v}^T\boldsymbol{x}_i$ and define $c_0 = c_1 - \epsilon$, $c_{K+1} = c_K + \epsilon$ for arbitrary $\epsilon > 0$. Then, we choose $\boldsymbol{W}_1, b_1$ as in Eq. 28, using $\boldsymbol{v}$ and $\epsilon$ defined above.

Here, we focus on the relationship between the outputs of layer 1, when the neural network inputs are $\boldsymbol{x}_5, \boldsymbol{x}_6, \boldsymbol{x}_8$, respectively. From the definition of $c_i$, we have $c_6 \in (c_5, c_8)$. Using Eq. 24 and Eq. 28, we have $\alpha_j^1(\boldsymbol{x}_6) \in (\alpha_j^1(\boldsymbol{x}_5), \alpha_j^1(\boldsymbol{x}_8))$, meaning that the input $\alpha_j^1$ of layer 1 (for $\boldsymbol{x}_6$) is bounded by $\alpha_j^1$ for $\boldsymbol{x}_5$ and $\alpha_j^1$ for $\boldsymbol{x}_8$. After passing it

through the ReLU activation, we also have

$$\beta_j^1(\boldsymbol{x}_6) \in (\beta_j^1(\boldsymbol{x}_5), \beta_j^1(\boldsymbol{x}_8)), \qquad (27)$$

for all $j \in \{1, \cdots, \sqrt{J}\}$ using the monotonicity of ReLU. Kore precisely, we have $\beta_1^1(x_6) = +1$, $-1 \leq \beta_2^1(x_6) \leq +1$, and $\beta_3^1(x_6) = -1$. This is illustrated in the first layer in Fig. 8. Now we move to the construction of the second layer. Recall that the main idea of constructing $\boldsymbol{W}_2, b_2$ is using the linear system in Eq. 33. Using Eq.32 and the fact that we can set all elements of $\boldsymbol{W}_2$ as positive (as shown in [Yun et al., 2019]), Eq. 27 implies $\alpha_j^2(\boldsymbol{x}_6) \in (\alpha_j^2(\boldsymbol{x}_5), \alpha_j^2(\boldsymbol{x}_8))$ for all $j \in \{1, \cdots, \sqrt{J}\}$. Finally, we set the activation function of layer 2 as

$$\sigma_L(t) = \begin{cases} t, & \text{if } |t| \leq 1, \\ 0, & \text{if } |t| \geq 1 \end{cases}$$

and arbitrarily increase $\lambda$ such that $\beta_j^2(\boldsymbol{x}_6) = 0$. Then, the output of the network $g_\theta(\boldsymbol{x}) = \sigma_L(\boldsymbol{W}_2(\sigma_H(\boldsymbol{W}_1\boldsymbol{x} + \boldsymbol{b}_1)) + \boldsymbol{b}_2)$ satisfies $g_\theta(\boldsymbol{x}_6) = 0$.

Now the question is, what is the upper bound on $|J|$? Recall that Lemma 4.7 guarantees that $|J| \leq 3N + 2$. Since $\sigma_H$ and $\sigma_L$ can be represented by 2 ReLUs and 4 ReLUs, respectively, $2\sqrt{3N+2} + 4\sqrt{3N+2} = 6\sqrt{3N+2}$ ReLU neurons are sufficient for our 3-layer network construction.

# 6 EXTENSION TO OTHER NEURAL NETWORKS

For deeper neural networks, a lower bound on $N$ can be obtained in terms of the maximum width $d$, the number of layers $L$, and the number of pre-trained samples $K$.

**Proposition 6.1.** *For $L \geq 4$, $K \geq 3$, $T \subseteq [K]$, $|T| = N$, there exists an $L$-layer ReLU network with maximum width $d$ satisfying equation 3 and*

$$d \leq 4 \min \left\{ \sqrt{\frac{3N}{\sqrt{\left\lfloor \frac{L-1}{2} \right\rfloor}} + 5}, \sqrt{K} \right\} + 2.$$

The main challenge for proving Proposition 6.1 lies in constructing a suitable neural network, which can be addressed by utilizing our 3-layer network from Section 5 and the construction idea provided in Figure 2 of [Yun et al., 2019]. See Appendix B for the proof of Proposition 6.1.

# 7 EXPERIMENTS

In this section, we provide experimental results on a synthetic dataset, which supports our theoretical results. The experimental setup is as follows. We first randomly generate $K$ samples $D = \{(\boldsymbol{x}_i, y_i)\}_{i=1}^K$ where the feature

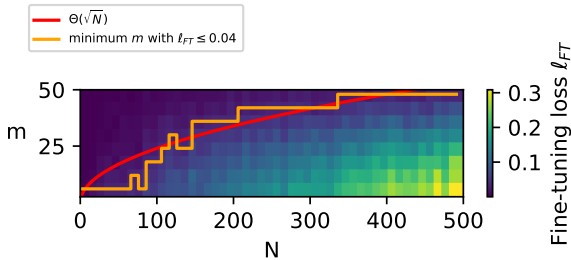

Figure 9: Fine-tuning loss $\ell_{\text{FT}}$ computed for a network trained on a synthetic dataset, for various $N$ and $m$. Here, the number of data points used for pre-training is set to $K = 1000$. As shown in the orange line, the number of neurons $m$ required in the fine-tuning network to achieve a small loss $\ell_{\text{FT}}$ grows in the order of $\Theta(\sqrt{N})$, the square root of the number of modified samples. This result coincides with our theoretical result in Theorem 5.1.

and the label of $i$-th sample have the following distributions: $\boldsymbol{x}_i \sim N(0, I_d)$ and $y_i \sim \text{Unif}[-1, 1]$, where the feature dimension is $d = 10$. Then, we train a network ReLU network $f$ that fits the dataset $D$, *i.e.,* equation 1 holds, thus having zero mean-squared-error (MSE) loss $\ell = \frac{1}{K} \sum_{i=1}^K (f(\boldsymbol{x}_i) - y_i)^2$. Considering the fine-tuning scenario, we construct another dataset $D' = \{(\boldsymbol{x}_i, y_i')\}_{i=1}^K$ as follows. We first initialize $D' = D$. Then, we randomly choose $N$ out of $K$ samples in $D'$, and re-define the label of the $N$ samples as $y_i' \sim \text{Unif}[-1, 1]$. Finally, we implement the fine-tuning process. Following our additive fine-tuning scenario, we freeze $f$ and train a 3-layer ReLU network $g_\theta$ with $m$ neurons, in a way that $f + g_\theta$ fits the new dataset $D'$. We define the fine-tuning loss as $\ell_{FT} = \frac{1}{K} \sum_{i=1}^K (f(\boldsymbol{x}_i) + g_\theta(\boldsymbol{x}_i) - y_i')^2$.

Figure 9 shows the fine-tuning loss $\ell_{FT}$ for different $m$ and $N$. As expected, for a given $N$, the fine-tuning loss decreases as $m$ increases. For each $N$, the yellow line in the figure shows the minimum $m$ satisfying $\ell_{FT}(m, N) \leq 0.04$. This yellow line indicates that the required number of neurons to achieve small fine-tuning loss follow the tendency of $\Theta(\sqrt{N})$ shown in the red line in the figure, which coincides with our theoretical result in Theorem 5.1.

# 8 CONCLUSION

We introduced Fine-Tuning Capacity (FTC), a generalization of memorization capacity concept for fine-tuning applications. This concept is defined to provide theoretical view on current paradigm of fine-tuning large pre-trained models. As an initial step towards analyzing FTC, we focused on the additive fine-tuning scenario where a side network is added to the frozen pre-trained network. We obtained upper/lower bounds on FTC for shallow ReLU networks. For 2-layer network, we showed that fine-tuning $N$ samples is possible by using ReLU networks with $m = \Theta(N)$ neurons, irrespective of the size of the pre-trained network and the number of total samples $K$ used during pre-training. For

3-layer network, the required amount of neurons reduces to $m = \Theta(\sqrt{N})$, for practical scenarios where the number of samples $N$ we want to change labels is far less than the number of total samples $K$ used for pre-training.

## Acknowledgements

JS acknowledges the support of the National Research Foundation of Korea (NRF) grant funded by the Korea government (MSIT) No. RS-2024-00345351. DK was partially supported by the National Research Foundation of Korea (NRF) grant funded by the Korea government (MSIT) (No. RS-2023-00252516) and the POSCO Science Fellowship of POSCO TJ Park Foundation. KL was supported by the NSF Award CCF-2339978 and a grant from FuriosaAI.

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

# Memorization Capacity for Additive Fine-Tuning
# (Supplementary Material)

Jy-yong Sohn[*1]     Dohyun Kwon[*2,3]     Seoyeon An[1]     Kangwook Lee[4]

[1]Department of Statistics and Data Science, Yonsei University, Republic of Korea
[2]Department of Mathematics, University of Seoul, Republic of Korea
[3]Center for AI and Natural Sciences, Korea Institute for Advanced Study, Republic of Korea
[4]Department of Electrical and Computer Engineering, University of Wisconsin-Madison, WI, USA

## A PROOF OF $m \leq 4N + 4$ (DEFERRED FROM SEC. 5.2)

We construct a 3-layer neural network $g_\theta$ with $4N + 4$ ReLU neurons which fine-tune $N$ samples. Note that we can find $\boldsymbol{v}$ such that $\boldsymbol{v}^T \boldsymbol{x}_i \neq \boldsymbol{v}^T \boldsymbol{x}_j$ for all $i \neq j$, since we assume $\boldsymbol{x}_i \neq \boldsymbol{x}_j$ for all $i \neq j$. Without loss of generality, we order samples such that $\boldsymbol{v}^T \boldsymbol{x}_1 < \boldsymbol{v}^T \boldsymbol{x}_2 < \cdots < \boldsymbol{v}^T \boldsymbol{x}_K$. Let $c_i := \boldsymbol{v}^T \boldsymbol{x}_i$ and define $c_0 = c_1 - \epsilon$, $c_{K+1} = c_K + \epsilon$ for arbitrary $\epsilon > 0$. Recall that the indices for the samples we want to fine-tune is denoted by $T = \{T_1, \cdots, T_N\}$. We define dummy indices $T_0 = 1$ and $T_{N+1} = K$. Consider $2N + 1$ groups of disjoint indices,

$$
\begin{aligned}
s_1 &= \{T_0, \cdots, T_1 - 1\}, \quad s_2 = \{T_1\}, \\
s_3 &= \{T_1 + 1, \cdots, T_2 - 1\}, \quad s_4 = \{T_2\}, \\
&\cdots, \quad s_{2N} = \{T_N\}, \\
s_{2N+1} &= \{T_N + 1, \cdots, T_{N+1}\},
\end{aligned}
$$

where group $s_1$ is empty when $T_1 = 1$, and group $s_{2N+1}$ is empty when $T_N = K$. We denote the maximum/minimum element of set $s_j$ as $s_j^{\max}$ and $s_j^{\min}$, respectively, i.e., $s_j^{\max} = \max s_j$ and $s_j^{\min} = \min s_j$. We place $2N + 1$ neurons on layer 1, and define parameters for layer 1 as

$$
\boldsymbol{W}_{1,j} = (-1)^{j-1} \frac{4}{c_{s_j^{\max}} + c_{s_{j+1}^{\min}} - c_{s_{j-1}^{\max}} - c_{s_j^{\min}}} \boldsymbol{v}^T \tag{28}
$$

$$
b_{1,j} = (-1)^j \frac{c_{s_j^{\max}} + c_{s_{j+1}^{\min}} + c_{s_{j-1}^{\max}} + c_{s_j^{\min}}}{c_{s_j^{\max}} + c_{s_{j+1}^{\min}} - c_{s_{j-1}^{\max}} - c_{s_j^{\min}}} \tag{29}
$$

for all $j = 1, \cdots, 2N + 1$. Under such setting, it can be easily checked that

$$
\begin{aligned}
\alpha_j^1(\boldsymbol{x}_i) \in [-1, 1] \quad &\text{for } i \in s_j \\
\alpha_j^1(\boldsymbol{x}_i) \notin [-1, 1] \quad &\text{for } i \notin s_j
\end{aligned}
$$

for all $j \in [2N + 1]$, where $\alpha_j^l(\boldsymbol{x})$ defined in Eq. 24 is the input value of node $j$ in layer $l$, when the input for the network is $\boldsymbol{x}$. Fig. 10 shows the example of such construction, when $K = 16$, $N = 2$ and $T = \{4, 7\}$. In such case, we have $2N + 1 = 5$ disjoint groups:

$$
\begin{aligned}
s_1 &= \{1, 2, 3\}, \quad s_2 = \{4\}, \quad s_3 = \{5, 6\}, \\
s_4 &= \{7\}, \quad s_5 = \{8, 9, \cdots, 16\}.
\end{aligned}
$$

As in Fig. 10, the input of $j$-th neuron lie in the non-clipping region $\alpha_1^j(\boldsymbol{x}_i) \in [-1, 1]$ when $i \in s_j$. For example, the first neuron ($j = 1$) in the first layer has $\alpha_1^1(\boldsymbol{x}_i) \in [-1, 1]$ for $i \in s_1 = \{1, 2, 3\}$.

---

[*]Equal Contribution

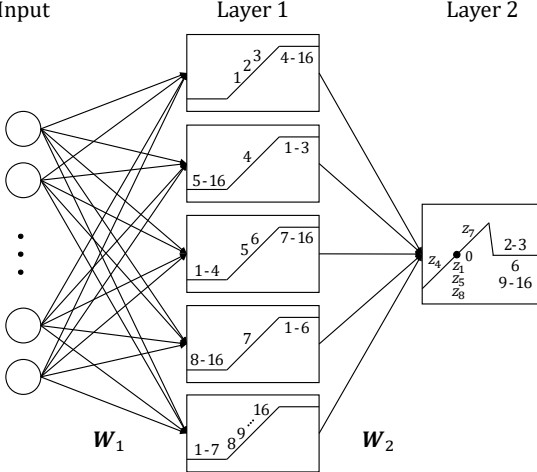

Figure 10: The construction of 3-layer network achieving upper bound $U_3 = 4N + 4$ in Sec. 5.2. Here, we consider the case when $K = 16$, $N = 2$ and $T = \{4, 7\}$.

Now we construct layer 2 as below. Let $S = \cup_{j=1}^{2N+1}\{s_j^{\min}\}$ be the index set containing the minimum index of each group $s_j$ for $j \in [2N+1]$, which is $S = \{1, 4, 5, 7, 8\}$ in the above example. Our goal is to construct $\boldsymbol{W}_2 \in \mathbb{R}^{2N+1}$ and $b_2 \in \mathbb{R}$ such that a single node in layer 2 locates (1) $\{\boldsymbol{x}_i\}_{i \in S}$ to the desired target $z_i$, and (2) $\{\boldsymbol{x}_i\}_{i \notin S}$ to the clipping region $[-1, 1]^c$. In other words, our desired conditions are

$$\alpha_1^2(\boldsymbol{x}_i) = z_i, \quad \forall i \in S, \tag{30}$$

$$\alpha_1^2(\boldsymbol{x}_i) \in [-1, 1]^c, \quad \forall i \notin S. \tag{31}$$

Note that the input of the first node in the second layer is represented as

$$\alpha_1^2(\boldsymbol{x}_i) = \sum_{j=1}^{2N+1} W_{2,j}\beta_j^1(\boldsymbol{x}_i) + b_2 \tag{32}$$

where $\boldsymbol{W}_2 = [W_{2,1}; \cdots; W_{2,2N+1}]$ and

$$\beta_j^1(\boldsymbol{x}_i) = \sigma_H(\alpha_j^1(\boldsymbol{x}_i))$$

is the output of node $j$ in layer 1, when the input to the network is $\boldsymbol{x}_i$. Thus, the first condition in Eq. 30 can be represented as a linear system

$$\boldsymbol{K}\begin{bmatrix} \boldsymbol{W}_2 \\ b_2 \end{bmatrix} = \begin{bmatrix} z_{i_1} \\ \vdots \\ z_{i_{2N+1}} \end{bmatrix} \tag{33}$$

where $i_k$ for $k \in [2N+1]$ is defined the as the elements of $S = \{i_1, \cdots, i_{2N+1}\}$ and

$$\boldsymbol{K} = \begin{bmatrix} \beta_1^1(\boldsymbol{x}_{i_1}) & \cdots & \beta_{2N+1}^1(\boldsymbol{x}_{i_1}) & 1 \\ \vdots & \ddots & \vdots & \vdots \\ \beta_1^1(\boldsymbol{x}_{i_{2N+1}}) & \cdots & \beta_{2N+1}^1(\boldsymbol{x}_{i_{2N+1}}) & 1 \end{bmatrix}. \tag{34}$$

In our above example, we have

$$\boldsymbol{K} = \begin{bmatrix} \beta_1^1(\boldsymbol{x}_1) & +1 & -1 & +1 & -1 & 1 \\ +1 & \beta_2^1(\boldsymbol{x}_4) & -1 & +1 & -1 & 1 \\ +1 & -1 & \beta_3^1(\boldsymbol{x}_5) & +1 & -1 & 1 \\ +1 & -1 & +1 & \beta_4^1(\boldsymbol{x}_7) & -1 & 1 \\ +1 & -1 & +1 & -1 & \beta_5^1(\boldsymbol{x}_8) & 1 \end{bmatrix}$$

Using a similar technique used in [Yun et al., 2019], this matrix $K \in \mathbb{R}^{(2N+1) \times (2N+2)}$ satisfies two conditions:

$$(1) \; \text{rank}(K) = 2N + 1,$$
$$(2) \; \exists \nu = [\nu_1, \cdots, \nu_{2N+2}] \in \text{null}(K)$$
$$\text{such that } \nu_i > 0 \quad \forall i \in [2N+1].$$

Thus, the linear system in Eq. 33 has infinitely many solution in the form of

$$\begin{bmatrix} W_2 \\ b_2 \end{bmatrix} = \mu + \lambda \nu \tag{35}$$

for any scalar $\lambda$ and a particular solution $\mu$. With the logic used in the proof of Lemma B.1 in [Yun et al., 2019], we can scale $\lambda$ sufficiently such that the second condition in Eq. 30 holds. Thus, by using such weight $W_2$ and bias $b_2$, the input of layer 2 looks like in Fig. 10. Let the neuron in layer 2 has activation function

$$\sigma_T(t) = \begin{cases} t, & \text{if } t < 1 \\ -\frac{1}{\delta}(t - 1 - \delta), & \text{if } 1 \le t < 1 + \delta \\ 0, & \text{if } t \ge 1 + \delta \end{cases}$$

Then, the output of the network

$$g_\theta(x) = \sigma_T(W_2(\sigma_H(W_1 x + b_1)) + b_2) = \sigma_T(\alpha_1^2(x))$$

satisfies

$$g_\theta(x_i) = \begin{cases} z_i, & i \in T \\ 0, & i \in [K] \setminus T \end{cases}$$

by using the definition of $\sigma_T$ and Eq. 30. Note that the first layer of this construction uses $2N + 1$ neurons with $\sigma_H$ activation, and the second layer uses 1 neuron with $\sigma_T$ activation. Since $\sigma_H$ and $\sigma_T$ can be converted into 2 ReLU neurons, our construction use total $4N + 4$ neurons, which completes the proof.

## B  PROOF OF PROPOSITION 6.1

Recall the 3-layer network illustrated in Figure 8. For given $T \subset [K]$, let us denote this 3-layer network satisfying equation 3 by $g_{\theta, T}$ which has maximum width $4 \min\{\sqrt{3|T|} + 2, \sqrt{K}\}$. In what follows, we construct an $L$-layer network based on $g_{\theta, T}$.

We partition $T$ into $\lfloor \frac{L-1}{2} \rfloor$ subsets: $T_1, T_2, \cdots, T_{\lfloor \frac{L-1}{2} \rfloor}$ satisfying $|T_i| \le |T| / \lfloor \frac{L-1}{2} \rfloor + 1$ for all $i$. It can be easily seen that $g_{\theta, T_1}(x) + g_{\theta, T_2}(x) + \cdots + g_{\theta, T_{\lfloor \frac{L-1}{2} \rfloor}}(x)$ satisfies equation 3. Using the construction idea provided in Figure 2 of [Yun et al., 2019], the above function can be represented as an $L$-layer network, which has a maximum width less than or equal to

$$\max_i \{4 \min\{\sqrt{3|T_i| + 2}, \sqrt{K}\} + 2\}, \tag{36}$$

and we conclude.