# OpenReview forum: "Memorization Capacity for Additive Fine-Tuning with Small ReLU Networks"
_auai.org/UAI/2024/Conference — UAI 2024 poster_

### Official Review · Reviewer_jrGa · 2024-03-16

**Q2-1 Originality-Novelty:** 2
**Q2-2 Correctness-Technical Quality:** 3
**Q2-5 Clarity Of Writing:** 4

**Q1 Summary And Contributions:**

This paper studies memorization capacity for ReLU networks up to 3 layers, under a special fine-tuning setting. The paper starts with motivating a fine-tuning setting with $M$ pre-training samples, and $N < M$ fine-tuning samples. The approach is to first reduce the problem into "Memorizing $N$ samples with arbitrary labels using a ReLU networks, and keeping all other $M-N$ samples to label zero". The authors then prove the size of the newly added network parameters is independent of $M$ for large enough $M$. Finally, they prove tight bounds on the number of required neurons for the fine-tuning network.

**Q2-3 Extent To Which Claims Are Supported By Evidence:**

4: Excellent: all claims are supported by very convincing evidence (in the form of comprehensive experimental evaluation, rigorous mathematical proofs, detailed (pseudo-)code, precise references, well-motivated and realistic assumptions) and the authors deliver what they promise.

**Q2-4 Reproducibility:**

4: Excellent: key resources (e.g. proofs, code, data) are available and key details (e.g. proof sketches, experimental setup) are comprehensively described for competent researchers to confidently and easily reproduce the main results.

**Q3 Main Strengths:**

* The paper is well-written and easy to follow.
* The problem of memorizing $M$ examples while keeping $N$ other examples intact is interesting, and the introduced bounds are tight for small ReLU networks.
* The introduced partitioning idea is interesting.

**Q4 Main Weakness:**

* The main weakness is the way the task is motivated. Firstly, the approach diverges from conventional fine-tuning in two significant aspects:
  * The authors assume network $f$ with frozen parameters $\theta$ is fixed, and seek a new side-tuning network $g$ with tunable parameters $\theta'$ for a final label generation of $y_i = f_\theta(x_i) + g(x_i)$. However, in fine-tuning (as well as LoRA), the goal is to find parameters $\theta'$ such that $y_i = f_{(\theta + \theta')}(x_i)$. The later is much different (and probably poses more challenges) than the former.
  * The motivation for "changing the label of a few examples while keeping the rest zero" is quite weak. While the authors cite reasons such as "adjustment of N labels", and "address task differences or to correct the pre-trained network’s errors", the reasons do not reflect how fine-tuning is used in practice.
* The majority of the proof technique is borrowed from [Yun et al., 2019].

**Q5 Detailed Comments To The Authors:**

I suggest the authors refine the motivation behind side-tuning. A few suggestions for such an improvement is as follows:
* The paper is entirely focused on shallow ReLU networks, I suggest the authors to somehow incorporate this in the title of the paper.
* The studied problem is more similar to side-tuning than fine-tuning. I suggest replacing fine-tuning with side-tuning to avoid any confusion with conventional fine-tuning (or parameter-efficient fine-tuning methods such as LoRA).
* The theorem statement can be improved. For instance, instead of proposing to "modify labels of $N$ examples", proposing to "Add $N$ examples without disturbing $M-N$ examples" could significantly improve the motivation behind your theoretical task. Although this is somehow mentioned in Remark 1, I suggest the authors incorporate this in their main theorem rather than a remark. Then, the studied task could have much stronger motivations, such as "continual learning".

Question: Is it possible to extend this proof technique to larger number of ReLU layers? For instance by further borrowing ideas from [Yun et al 2019] and introducing assumptions like general position?

**Q9 Complying With Reviewing Instructions:**

Yes

---

> ### Author Rebuttal · Authors · 2024-04-05
>
> We thank Reviewer jrGa for the helpful review. We are glad **the reviewer finds the problem formulation and the novel partitioning idea interesting.**
>
> **``[R5-1]. The additive fine-tuning approach diverges from conventional fine-tuning``**
>
> The reviewer is correct that popular fine-tuning methods do not match our additive model. For the ease of analysis, we started with a simple model of additive fine-tuning [Zhang'20, Fu'21, Cao'22], and extending our result to popular fine-tuning is considered as a future work.
>
> We revised the introduction to clarify the current status and future direction of our work.
>
> > **<Revised paragraph in Sec.1>**
> >
> > …This scenario is motivated by recently proposed additive fine-tuning methods [Zhang'20, Fu'21, Cao'22], and especially, the side-tuning [Zhang'20] where a side network $g_{\theta}$ is added to the pre-trained network $f$. Since our model does not cover other popular fine-tuning methods including LoRA [Hu'21], extending our theoretical results to such popular methods is remained as a future work.
>
> [Zhang'20] Zhang, Jeffrey O., et al. "Side-tuning: a baseline for network adaptation via additive side networks." ECCV 2020.
>
> [Fu'21] Fu, Cheng, et al. "Learn-to-share: A hardware-friendly transfer learning framework exploiting computation and parameter sharing." ICML 2021.
>
> [Hu'21] Edward J Hu, Yelong Shen, Phillip Wallis, Zeyuan Allen- Zhu, Yuanzhi Li, Shean Wang, Lu Wang, and Weizhu Chen. "Lora: Low-rank adaptation of large language models", ICLR 2022.
>
> [Cao'22] Cao, Jin, Chandana Satya Prakash, and Wael Hamza. "Attention Fusion: a light yet efficient late fusion mechanism for task adaptation in NLU." NAACL 2022.
>
>
> **`[R5-2]. Proof technique mostly borrowed from [Yun'19]`**
>
> The reviewer is right that we mostly inherited the proof techniques used in [Yun'19], but we added small modification to get our upper bound result; to satisfy that $(M-N)$ labels are maintained before/after the fine-tuning, we introduced new tricks (e.g., counting the cardinality of the consecutive partition in Sec.4.2) for the network construction.
> We emphasize that the main novelty of this paper is about a new *problem formulation* for the fine-tuning scenario, not about a new proof technique.
>
>
> **``[R5-3]. Change the title to clarify that only shallow ReLU networks are considered``**
>
> As per the reviewer's comment, we will change the title to "Memorization Capacity for Additive Fine-Tuning with Small ReLUs".
>
>
>
> **``[R5-4]. Refine the motivation behind the problem formulation``**
>
> As per the reviewer's comment, we refined the motivation behind current problem setup as below.
>
> > **[Refined Problem Setup (for better motivation)]**
> > Let $f$ be an arbitrary pre-trained network, and $D=\{(x_i, y_i)\}_{i=1}^M$ be the dataset we use for fine-tuning.
> >
> > Our goal is to find a neural network $g_{\theta}$ such that $(f+g_{\theta})(x_i) = y_i$ for all $i \in [M]$. Let $T$ be the set of sample indices $i \in [M]$ satisfying $f(x_i) \ne y_i$, i.e., the pre-trained network $f$ does not fit, and let $N=|T|$. We want to find the minimum number of neurons $m$ required in $g_{\theta}$ for successful fine-tuning.
>
> This refined problem setup aligns with the reviewer's suggestion of interpreting it as continual learning setup of "adding $N$ examples without disturbing $M-N$ examples".
>
> **``[R5-5]. Possible to extend the result to deeper ReLU networks?``**
>
> Solving this problem is interesting, but non-trivial. We established two conjectures on theoretical results for deeper ReLU networks with $L \ge 4$ layers.
>
> > **Conjecture 1 (Relationship between fine-tuning capacity and memorization capacity for general $L$)**: Let $g_{\theta}$ be a ReLU network with $m$ neurons and $L$ layers, and let $M$ be the number of pre-training samples. If $M \gg m$, the fine-tuning capacity $N$ of $g_{\theta}$ is given by:
> >
> >$$N \approx \frac{\text{memorization capacity of } g_{\theta}}{3}.$$
>
>
> > **Conjecture 2 (Bound on fine-tuning capacity for general $L$)**: Let $M$ be the number of pre-training samples, and $N$ be the fine-tuning capacity. There exists an $L$-layer ReLU network with maximum width $d$ satisfying the fine-tuning objective and  $$ d \leq   \frac{6}{\sqrt{\left\lfloor\frac{L-1}{2}\right\rfloor}}\min\{\sqrt{3N+2}, \sqrt{M}\} + 1.$$
>
> The conjecture 2 can be proved by constructing an appropriate neural network, by utilizing our 3-layer network from Section 5 and the construction idea provided in Figure 2 of [Yun'19].
>
> In the revised manuscript, we will add the detailed proof of conjecture 2, and state that a lower bound on fine-tuning capacity (FTC) $N$ is provided for general $L$. Proving conjecture 1 remains open for $L \geq 4$, which we will highlight as a future research direction.
>
>
> [Yun'19] Yun, Chulhee, Suvrit Sra, and Ali Jadbabaie. "Small relu networks are powerful memorizers: a tight analysis of memorization capacity". NeurIPS 2019.

---

### Official Review · Reviewer_nDxV · 2024-03-21

**Q2-1 Originality-Novelty:** 1
**Q2-2 Correctness-Technical Quality:** 3
**Q2-5 Clarity Of Writing:** 2

**Q1 Summary And Contributions:**

This paper considers the memorization capacity of ReLU neural networks in a certain case of additive fine-tuning. More precisely, this paper studies the number of neurons required for memorizing $N$ samples with 2/3-layer ReLU neural networks while having the constraint to be zero on the other $M-N$ samples. (The form of fine-tuning considered in the paper is summing a small network to the original pretrained network.) Interestingly, it is shown that as the number of zero samples $M-N$ increases, we don't need to use larger ReLU networks for memorizing them.

**Q2-3 Extent To Which Claims Are Supported By Evidence:**

4: Excellent: all claims are supported by very convincing evidence (in the form of comprehensive experimental evaluation, rigorous mathematical proofs, detailed (pseudo-)code, precise references, well-motivated and realistic assumptions) and the authors deliver what they promise.

**Q2-4 Reproducibility:**

4: Excellent: key resources (e.g. proofs, code, data) are available and key details (e.g. proof sketches, experimental setup) are comprehensively described for competent researchers to confidently and easily reproduce the main results.

**Q3 Main Strengths:**

- Interestingly, it is shown that no matter how large the number of samples we'd like to keep unchanged is, the size of the required network would mainly depend on the number of samples we'd like to fine-tune on.
- The proofs are rather insightful by showing that we can make ReLU networks fit by being almost always constant and having a few bumps.

**Q4 Main Weakness:**

- Personally, I think the problem formulation can be better motivated. Particularly, why consider this certain format of additive fine-tuning and requiring most of the labels to be fixed and changing the output on the rest of the inputs? E.g., in many of the fine-tuning routines, a new data set is used for fine-tuning and the change of the neural network's output on other values is not really a concern. So it's mostly like memorizing new samples from scratch. The pertaining can hugely affect the learning part on the fine-tuning side through learning dynamics, but I'm not fully convinced if the connection between pre-training and memorization as made in this paper is relevant.

**Q5 Detailed Comments To The Authors:**

The paper's presentation can be improved. Other than the points mentioned in weaknesses:

- 1. Definition 1 is not complete on its own and requires the reader to remember the aforementioned settings. I believe, this can be improved and definitions and theorems should be as self-contained as possible, and when not possible, with appropriate references to the text.
- 2. $m(N)$ is not defined in Thm 1.2.
- Figure 4 can be improved (i.e., saying that $N$ is fixed and $M$ is varied and we're looking into the memorization capacity). Also the condition $M\geq N$ is not reflected in this figure.

**Q9 Complying With Reviewing Instructions:**

Yes

---

> ### Author Rebuttal · Authors · 2024-04-05
>
> We thank  Reviewer nDxV for the detailed review and very helpful suggestions. We appreciate that Reviewer nDxV finds that **the theoretical result in our paper is interesting, and the corresponding proofs are insightful**. Below we handle the comment received from the reviewer.
>
> **`[R4-1]. The problem formulation is less motivating, since the additive fine-tuning scenario is different from many practical fine-tuning routines`**
>
> Thanks for the insightful question. We agree that the additive fine-tuning scenario has some gap with popular fine-tuning methods including LoRA. At the same time, we argue that for the sake of theoretical analysis, we considered the simplest mathematical model that reflects one fine-tuning method called side-tuning [Zhang'20]. Although our current additive model does not cover many fine-tuning methods, we expect our pioneering work provides a meaningful direction towards theoretically understanding the capability of fine-tuning a pre-trained model.
>
> In addition, we want to emphasize that our theoretical results have practical implications. For example, in our response `[R1-2]` to `Reviewer EjGX`, we show our experimental results showing that we can achieve small enough fine-tuning loss when the number of neurons $m$ is in the order of $\Theta(\sqrt{N})$, which coincides with our theoretical result in Theorem 5.1. This shows that our theory provides a guideline on how many neurons a neural network need to contain, for the purpose of successful fine-tuning.
>
> In the revised manuscript, we will include this discussion and experimental results.
>
>
> [Zhang'20] Zhang, Jeffrey O., et al. "Side-tuning: a baseline for network adaptation via additive side networks." Computer Vision–ECCV 2020: 16th European Conference, Glasgow, UK, August 23–28, 2020, Proceedings, Part III 16. Springer International Publishing, 2020.
>
>
> **`[R4-2]. Improve the presentation (Def.1, Thm.1.2, Fig.4)`**
>
> Thanks for the constructive comments. In the revised manuscript, we will make things clear:
> * The reviewer is correct that Definition 1 is not self-contained. We will restate the aformentioned settings within Definition 1.
> * The notation $m=m(N)$ is used to emphasize that the minimum number of neurons $m$ depends on $N$. The specific formula for $m(N)$ are included in Theorem 4.1 and Theorem 5.1. We will clearly state this in Theorem 1.2.
> * Following the reviewer's suggestion, we will revise Figure 4 as below to clearly state the setting:
> 1. $N$ is fixed, and we plot the minimum number of neurons $m$ required for fine-tuning or memorization, as $M$ varies.
> 2. The valid regime in the horizontal axis is $M\ge N$
>
> **`[R4-3]. Proof idea is very close to prior work`**
>
> The reviewer is right that we mostly inherited the proof techniques used in [Yun'19], but we added small modification to get our upper bound result; to satisfy that $(M-N)$ labels are maintained before/after the fine-tuning, we introduced new tricks (e.g., counting the cardinality of the consecutive partition in Sec.4.2) for the network construction.
>
> We emphasize that the main novelty of this paper is about a new *problem formulation* for the fine-tuning scenario, not about a new proof technique.
>
> [Yun'19] Yun, Chulhee, Suvrit Sra, and Ali Jadbabaie. "Small relu networks are powerful memorizers: a tight analysis of memorization capacity". NeurIPS 2019.

---

### Official Review · Reviewer_nuq4 · 2024-03-23

**Q2-1 Originality-Novelty:** 2
**Q2-2 Correctness-Technical Quality:** 3
**Q2-5 Clarity Of Writing:** 3

**Q1 Summary And Contributions:**

This paper studies the minimum number of neurons required to fine-tune labels among a certain sample size, alternatively identifying the maximum number of labels a network can fine-tune. The research specifically demonstrates that for 2-layer networks, m is directly proportional to $N$, meaning $m=\Theta(N)$ neurons are adequate for fine-tuning N samples. On the other hand, for 3-layer networks, $m$ is directly proportional to the square root of N, i.e., $m=\Theta(\sqrt N)$, is sufficient for fine-tuning the same number of samples.

**Q2-3 Extent To Which Claims Are Supported By Evidence:**

3: Good: the main claims are supported by convincing evidence (in the form of adequate experimental evaluation, proofs, (pseudo-)code, references, assumptions).

**Q2-4 Reproducibility:**

3: Good: key resources (e.g. proofs, code, data) are available and key details (e.g. proofs, experimental setup) are sufficiently well-described for competent researchers to confidently reproduce the main results.

**Q3 Main Strengths:**

This paper extends traditional memorization capacity, which studies the maximum number of labels that can be represented, to fine-tuning capacity defined as the maximum number of labels that can be fine-tuned. This research novelly clusters the 'neighbors' of identical labels to a singular point, thereby reducing the number of linear pieces dependent solely on the quantity of fine-tuned labels $N$. As a result, a clear distinction is established between the memorization of all $M$ samples and the fine-tuning of just N samples.

**Q4 Main Weakness:**

This paper considers only a specific setting, in which the fine-tuning is viewed as a summation of pre-trained network and another network. Also, most results seem a direct extension of previous results of memorization capacity.

**Q5 Detailed Comments To The Authors:**

For 3-layer networks, the authors have claimed that the number of required neurons is $m=\Theta(\sqrt N)$. However, the justification given in Remark 5 doesn't seem quite clear. Also, given the lower/upper bound of $\Theta(\sqrt N)$, other results in Theorem 5.1 seem somehow incremental. Do these results show more insight to understand or make the bound tighter?

**Q9 Complying With Reviewing Instructions:**

Yes

---

> ### Author Rebuttal · Authors · 2024-04-05
>
> We thank Reviewer nuq4 for the helpful review. We are glad the reviewer sees a clear distinction between the proposed fine-tuning capacity and memorization capacity.
>
> **``[R3-1]. This paper only focuses on the fine-tuning setting where another network is added to the pre-trained network``**
>
> The reviewer is correct that popular fine-tuning methods do not match our additive model. For the ease of analysis, we started with a simple model of additive fine-tuning [Zhang'20, Fu'21, Cao'22], and extending our result to popular fine-tuning is considered as a future work.
>
> We revised the introduction to clarify the current status and future direction of our work.
>
> > **<Revised paragraph in Sec.1>**
> >
> > …This scenario is motivated by recently proposed additive fine-tuning methods [Zhang'20, Fu'21, Cao'22], and especially, the side-tuning [Zhang'20] where a side network $g_{\theta}$ is added to the pre-trained network $f$. Since our model does not cover other popular fine-tuning methods including LoRA [Hu'21], extending our theoretical results to such popular methods is remained as a future work.
>
> [Zhang'20] Zhang, Jeffrey O., et al. "Side-tuning: a baseline for network adaptation via additive side networks." ECCV 2020.
>
> [Fu'21] Fu, Cheng, et al. "Learn-to-share: A hardware-friendly transfer learning framework exploiting computation and parameter sharing." ICML 2021.
>
> [Hu'21] Edward J Hu, Yelong Shen, Phillip Wallis, Zeyuan Allen- Zhu, Yuanzhi Li, Shean Wang, Lu Wang, and Weizhu Chen. "Lora: Low-rank adaptation of large language models", ICLR 2022.
>
> [Cao'22] Cao, Jin, Chandana Satya Prakash, and Wael Hamza. "Attention Fusion: a light yet efficient late fusion mechanism for task adaptation in NLU." NAACL 2022.
>
>
>
> **``[R3-2]. Most results seems a direct extension of previous results of memorization capacity.``**
>
> The reviewer is right that we mostly inherited the proof techniques used in [Yun'19], but we added small modification to get our upper bound result; to satisfy that $(M-N)$ labels are maintained before/after the fine-tuning, we introduced new tricks (e.g., counting the cardinality of the consecutive partition in Sec.4.2) for the network construction.
> We emphasize that the main novelty of this paper is about a new *problem formulation* for the fine-tuning scenario, not about a new proof technique.
>
> [Yun'19] Yun, Chulhee, Suvrit Sra, and Ali Jadbabaie. "Small relu networks are powerful memorizers: a tight analysis of memorization capacity". NeurIPS 2019.
>
>
> **`[R3-3]. For 3-layer networks, given that the authors have claimed` $m = \Theta(\sqrt{N})$, `the justification in Remark 5 is not clear.`**
>
>
> Thanks for pointing this out. We found a typo in Remark 5. It should say:
> "if $N=\Theta(\sqrt{M})$, then $m=\Theta(\sqrt[4]{M})$”, which can be directly obtained from Theorem 5.1, by plugging in $N=\Theta(\sqrt{M})$ in the equation for the upper bound on $m$.
>
> We revised Remark 5 to clarify that $m=\Theta(\sqrt{N})$:
> > **Remark 5 (revised version)**
> > * The upper bound in Theorem 5.1 directly shows that $m \le \Theta(\sqrt{N})$.
> > * The lower bound in Theorem 5.1 indicates two facts:
> > (1) If $3N \le M-2$, then $m \ge \Theta(\sqrt{N})$, and
> > (2) If $3N > M-2$, then $m \ge \Theta(\sqrt{M})$. Combining this with $M \ge N$, we have $m \ge \Theta(\sqrt{N})$.
> > * All in all, $m = \Theta(\sqrt{N})$.
>
> **`[R3-4]. Given the upper/lower bound in` $\Theta(\sqrt{N})$ `format, some results in Theorem 5.1 seem incremental.`**
>
> Recall that the upper bound in Theorem 5.1 has two parts:
> (1) $m \le 6\sqrt{3N+2}$, and
> (2) $m \le 2\sqrt{M} + \min (2\sqrt{M}, 3N)$.
>
> While (1) alone proves $m=\Theta(\sqrt{N})$, we expect (2) plays a role when rephrasing the theorem to bound $N$ instead of $m$, similar to how Corollary 4.2 bounds $N$ based on Theorem 4.1. The $M$-dependence in (2) matters for this.

---

### Official Review · Reviewer_1g7b · 2024-03-23

**Q2-1 Originality-Novelty:** 3
**Q2-2 Correctness-Technical Quality:** 3
**Q2-5 Clarity Of Writing:** 3

**Q1 Summary And Contributions:**

The paper proposes a novel metric known as Fine-tuning Capacity (FTC), designed to quantify the maximum number of samples a neural network can fine-tune. The main contributions are: 1) The establishment of a new definition for FTC, offering a generalization of the memorization capacity concept introduced by Yun et al., 2019. 2) The determination of precise upper and lower bounds for the minimum number of neurons $m$ required to arbitrarily modify $N$ labels among $M$ samples for 2- and 3-layer ReLU networks.

**Q2-3 Extent To Which Claims Are Supported By Evidence:**

3: Good: the main claims are supported by convincing evidence (in the form of adequate experimental evaluation, proofs, (pseudo-)code, references, assumptions).

**Q2-4 Reproducibility:**

3: Good: key resources (e.g. proofs, code, data) are available and key details (e.g. proofs, experimental setup) are sufficiently well-described for competent researchers to confidently reproduce the main results.

**Q3 Main Strengths:**

The paper is well-written and easy to follow as far as I concerned. The analysis comprehensively covers the upper and lower bounds of two cases:  2-layer and 3-layer ReLU networks. I appreciate the specific instance where $M=N$, demonstrating that the FTC results align with those presented by Zhang et al., 2016. This alignment underscores FTC's potential as an extension of the concept of memorization capacity.

**Q4 Main Weakness:**

The theoretical analysis is limited to shallow ReLU networks, specifically those with 2 or 3 layers. Is it possible to extend these results to networks with more layers or to different types of networks?

**Q5 Detailed Comments To The Authors:**

See weakness.

**Q9 Complying With Reviewing Instructions:**

Yes

---

> ### Author Rebuttal · Authors · 2024-04-05
>
> We thank  Reviewer 1g7b for the detailed review and very helpful suggestions. We appreciate that Reviewer 1g7b finds that **our paper is well-written and easy to follow, and fine-tuning capacity defined by us is a potential extension of memorization capacity**. Below we handle the comment received from the reviewer.
>
> **`[R2-1]. Is it possible to extend the theoretical results with more layers or to different types of networks?`**
>
>
> Thank you for your insightful comment on extending our theoretical results to more complex neural networks. We agree that this is an important research direction, but it is non-trivial and left as a future direction in our work.
>
> Below we share two conjectures we have (based on our observations and proof techniques), regarding extension of our result to $L$-layer ReLU network for arbitrary $L$.
>
>
> > **Conjecture 1 (Relationship between fine-tuning capacity and memorization capacity for general $L$)**: Let $g_{\theta}$ be a ReLU network with $m$ neurons and $L$ layers, and let $M$ be the number of pre-training samples. If $M \gg m$, the fine-tuning capacity $N$ of $g_{\theta}$ is given by:
> >
> >$$N \approx \frac{\text{memorization capacity of } g_{\theta}}{3}.$$
>
>
> > **Conjecture 2 (Bound on fine-tuning capacity for general $L$)**: Let $M$ be the number of pre-training samples, and $N$ be the fine-tuning capacity. There exists an $L$-layer ReLU network with maximum width $d$ satisfying the fine-tuning objective and  $$ d \leq   \frac{6}{\sqrt{\left\lfloor\frac{L-1}{2}\right\rfloor}}\min(\sqrt{3N+2}, \sqrt{M}) + 1.$$
>
> Note that conjecture 2 gives a lower bound on $N$ in terms of the maximum width $d$, the number of layers $L$, and the number of pre-trained samples $M$. We also note that without the restriction on the maximum width, the bound is indeed similar to the ones we have in Theorem 5.1, for 3-layer networks. The main challenge for proving conjecture 2 lies in constructing an appropriate neural network, which can be addressed by utilizing our 3-layer network from Section 5 and the construction idea provided in Figure 2 of [Yun'19].
>
> In the revised manuscript, we will add the detailed proof of conjecture 2, and state that a lower bound on fine-tuning capacity (FTC) $N$ is provided for general $L$. However, due to the gap between upper and lower bounds on the memorization capacity for general $L$, the equivalence between fine-tuning capacity and memorization capacity (stated in conjecture 1) remains open for $L \geq 4$, which we will highlight as a future research direction.
>
>
> [Yun'19] Yun, Chulhee, Suvrit Sra, and Ali Jadbabaie. "Small relu networks are powerful memorizers: a tight analysis of memorization capacity". NeurIPS 2019.

---

### Official Review · Reviewer_EjGX · 2024-03-25

**Q2-1 Originality-Novelty:** 2
**Q2-2 Correctness-Technical Quality:** 3
**Q2-5 Clarity Of Writing:** 3

**Q1 Summary And Contributions:**

This paper proposes a new measure, Fine-Tuning Capacity (FTC), to define the maximum number of samples a neural network can fine-tune, which extends the memorization capacity concept to the fine-tuning scenario. This paper also analyzes FTC for the additive fine-tuning scenario, which could be potentially useful for the theoretical analysis of some popular fine-tuning algorithms such as Lora.

**Q2-3 Extent To Which Claims Are Supported By Evidence:**

3: Good: the main claims are supported by convincing evidence (in the form of adequate experimental evaluation, proofs, (pseudo-)code, references, assumptions).

**Q2-4 Reproducibility:**

2: Fair: key resources (e.g. proofs, code, data) are unavailable but key details (e.g. proof sketches, experimental setup) are sufficiently well-described for an expert to confidently reproduce the main results.

**Q3 Main Strengths:**

1. This paper proposes a new measure, Fine-Tuning Capacity (FTC), to define the maximum number of samples a neural network can fine-tune, which extends the memorization capacity concept to the fine-tuning scenario.
2. This paper also analyzes FTC for the additive fine-tuning scenario, which could be potentially useful for the theoretical analysis of some popular fine-tuning algorithms such as Lora.

**Q4 Main Weakness:**

1. The theoretical analysis is limited to a ReLU network with either 2 or 3 layers, which is far from the complex neural networks used in practices. It is unknown whether the theoretical results from this paper could be extended to more complicated models.

2. There seems to be no experiments to verify the theoretical results.

**Q5 Detailed Comments To The Authors:**

1. Is it possible to design some experiments to verify whether the theoretical analysis matches the results in practice?

**Q9 Complying With Reviewing Instructions:**

Yes

---

> ### Author Rebuttal · Authors · 2024-04-05
>
> We thank the Reviewer EjGX for the detailed review and constructive suggestions. We appreciate your acknowledgments that **the theoretical analysis is novel, and potentially useful for other fine-tuning methods**. Please find our answers to your comments and questions as follows.
>
> **`[R1-1]. Can we extend current result to more complicated models, other than 2 or 3-layer ReLU networks?`**
>
> Thank you for your insightful comment on extending our theoretical results to more complex neural networks. We agree that this is an important research direction, but it is non-trivial and left as a future direction in our work.
>
> Below we share two conjectures we have (based on our observations and proof techniques), regarding extension of our result to $L$-layer ReLU network for arbitrary $L$.
>
>
> > **Conjecture 1 (Relationship between fine-tuning capacity and memorization capacity for general $L$)**: Let $g_{\theta}$ be a ReLU network with $m$ neurons and $L$ layers, and let $M$ be the number of pre-training samples. If $M \gg m$, the fine-tuning capacity $N$ of $g_{\theta}$ is given by:
> >
> >$$N \approx \frac{\text{memorization capacity of } g_{\theta}}{3}.$$
>
>
> > **Conjecture 2 (Bound on fine-tuning capacity for general $L$)**: Let $M$ be the number of pre-training samples, and $N$ be the fine-tuning capacity. There exists an $L$-layer ReLU network with maximum width $d$ satisfying the fine-tuning objective and  $$ d \leq   \frac{6}{\sqrt{\left\lfloor\frac{L-1}{2}\right\rfloor}}\min(\sqrt{3N+2}, \sqrt{M}) + 1.$$
>
> Note that conjecture 2 gives a lower bound on $N$ in terms of the maximum width $d$, the number of layers $L$, and the number of pre-trained samples $M$. We also note that without the restriction on the maximum width, the bound is indeed similar to the ones we have in Theorem 5.1, for 3-layer networks. The main challenge for proving conjecture 2 lies in constructing an appropriate neural network, which can be addressed by utilizing our 3-layer network from Section 5 and the construction idea provided in Figure 2 of [Yun'19].
>
> In the revised manuscript, we will add the detailed proof of conjecture 2, and state that a lower bound on fine-tuning capacity (FTC) $N$ is provided for general $L$. However, due to the gap between upper and lower bounds on the memorization capacity for general $L$, the equivalence between fine-tuning capacity and memorization capacity (stated in conjecture 1) remains open for $L \geq 4$, which we will highlight as a future research direction.
>
>
> [Yun'19] Yun, Chulhee, Suvrit Sra, and Ali Jadbabaie. "Small relu networks are powerful memorizers: a tight analysis of memorization capacity". NeurIPS 2019.
>
>
> **`[R1-2]. No experiments to verify the theoretical results?`**
>
>
> We thank the reviewer for the constructive comment. Following the reviewer's suggestion, we tested whether our theoretical results align with experimental observations.
>
>
> **Data setting:** We randomly generated $M$ samples $D=\{(x_i, y_i)\}_{i=1}^M$ where the feature and the label of $i$-th sample have the following distributions: $x_i \sim N(0, I_d)$ and $y_i \sim \text{Unif}[-1, 1]$, where the feature dimension is $d=10$.
>
> **Pre-training:** We trained a ReLU network $f$ to fit the dataset $D$, thus having zero mean-squared-error (MSE) loss $\ell = \frac{1}{M} \sum_{i=1}^M (f(x_i)-y_i)^2$.
>
> **Fine-tuning:** We construct a new dataset $D'=\{(x_i, y_i' )\}_{i=1}^M$ as follows. We first initialize $D' = D$. Then, we randomly choose $N$ out of $M$ samples in $D'$, and re-define the label of the $N$ samples as $y_i' \sim \text{Unif}[-1, 1]$.
>
> The fine-tuning process is defined as below: we freeze $f$ and train a 3-layer ReLU network $g_{\theta}$ with $m$ neurons, in a way that $f+g_{\theta}$ fits the new dataset $D'$. We define the fine-tuning loss as  $\ell_{FT} = \frac{1}{M}\sum_{i=1}^M (f(x_i) + g_{\theta}(x_i)-y_i')^2$.
>
> The figure in the link (https://hackmd.io/_uploads/SJFVOET1C.png) shows the fine-tuning loss $\ell_{FT}$ for different $m$ and $N$. As expected, for a given $N$ (number of new sample we need to fit), the fine-tuning loss decreases as $m$ increases. For each $N$, the yellow line in the figure shows the minimum $m$ satisfying $\ell_{FT}(m,N) \le 0.15$. This yellow line indicates that the required number of neurons for small enough fine-tuning loss (say 0.15) is similar to $\Theta(\sqrt{N})$ (shown as the red line in the figure), which coincides with our theoretical result in Theorem 5.1.
>
> In the revised manuscript, we will include this experimental result and its implication.

---

### Meta-Review · Area_Chair_PAp2 · 2024-04-15

The paper discussed ine-Tuning Capacity (FTC) for ReLU networks, extending the existing framework of memorization capacity to encompass fine-tuning scenarios.  The authors establish theoretical bounds for FTC and explore its implications through a combination of theoretical analysis and experimental validation.

The reviewers appreciate the clarity of the exposition and the rigorous mathematical analysis provided. The use of additive models for fine-tuning, while different from more conventional approaches, is acknowledged as a valid simplification for theoretical investigation. The conjectures proposed for extending the results to networks with more layers add depth to the discussion and outline clear directions for future research.

While there are criticisms regarding the practical relevance of the specific fine-tuning setup used, the authors seem to have effectively address these concerns in their rebuttal by explaining the choice of their model and pointing towards future extensions that could bridge the gap to more conventional fine-tuning methods. The experimental results provided support the theoretical claimxs. The main strength of the paper has been found to lie in its novel approach to quantifying fine-tuning ability and the potential implications this has for understanding the limits of network adaptability. This may impact how practitioners think about the capacity requirements of fine-tuning in various applications.